# *Sox9* and *Sox8* protect the adult testis from male-to-female genetic reprogramming and complete degeneration

Francisco J Barrionuevo[1,2][*][†], Alicia Hurtado[1,2][†], Gwang-Jin Kim[3][†], Francisca M Real[1,2][‡], Mohammed Bakkali[4], Janel L Kopp[5][§], Maike Sander[5], Gerd Scherer[3], Miguel Burgos[1,2], Rafael Jiménez[1,2][*]

[1]Departamento de Genética e Instituto de Biotecnología, Universidad de Granada, Granada, Spain; [2]Centro de Investigación Biomédica, Universidad de Granada, Granada, Spain; [3]Institute of Human Genetics, University of Freiburg, Freiburg, Germany; [4]Departamento de Genética, Facultad de Ciencias, Universidad de Granada, Granada, Spain; [5]Department of Pediatrics and Cellular and Molecular Medicine, University of California, San Diego, San Diego, United States

*For correspondence: fjbarrio@ugr.es (FJB); rjimenez@ugr.es (RJ)

[†]These authors contributed equally to this work

Present address: [‡]Max Planck Institute for Molecular Genetics, Berlin, Germany; [§]Department of Cellular and Physiological Sciences, University of British Columbia, Vancouver, Canada

Competing interests: The authors declare that no competing interests exist.

**Abstract** The new concept of mammalian sex maintenance establishes that particular key genes must remain active in the differentiated gonads to avoid genetic sex reprogramming, as described in adult ovaries after *Foxl2* ablation. *Dmrt1* plays a similar role in postnatal testes, but the mechanism of adult testis maintenance remains mostly unknown. *Sox9* and *Sox8* are required for postnatal male fertility, but their role in the adult testis has not been investigated. Here we show that after ablation of *Sox9* in Sertoli cells of adult, fertile *Sox8*[-/-] mice, testis-to-ovary genetic reprogramming occurs and Sertoli cells transdifferentiate into granulosa-like cells. The process of testis regression culminates in complete degeneration of the seminiferous tubules, which become acellular, empty spaces among the extant Leydig cells. DMRT1 protein only remains in non-mutant cells, showing that SOX9/8 maintain *Dmrt1* expression in the adult testis. Also, *Sox9/8* warrant testis integrity by controlling the expression of structural proteins and protecting Sertoli cells from early apoptosis. Concluding, this study shows that, in addition to its crucial role in testis development, *Sox9*, together with *Sox8* and coordinately with *Dmrt1*, also controls adult testis maintenance.

## Introduction

*Sox* genes encode an important group of transcription factors with relevant roles in many aspects of pre- and post-natal development of vertebrates and other animal taxa. There are 20 *Sox* genes in vertebrates, which are classified into 9 groups. *Sox8, Sox9*, and *Sox10 (SoxE* group) are involved in many developmental processes (reviewed in *Lefebvre et al., 2007*). All three *SoxE* genes are expressed during testis development, *Sox9* being essential for testis determination and *Sox9/Sox8* necessary for subsequent embryonic differentiation (*Chaboissier, 2004*, *Barrionuevo et al., 2006*, *Barrionuevo et al., 2009*). *Sox10* can substitute for *Sox9* during testis determination (*Polanco et al., 2010*). Undifferentiated gonads have the inherent potential to develop into two completely different organs, either as testes or as ovaries. The decision as to which fate to follow depends on the presence/absence of sex-specific factors. In the male, the Y-linked, mammalian sex-determining factor, *SRY*, upregulates *SOX9* which triggers testis differentiation, whereas in the female, the WNT/β -catenin

**eLife digest** Scientists thought for years that the ovaries and testes are fully developed, stable organs that cannot change their structure and function in mature mammals. However, more recent studies have shown that a gene called *Foxl2* is active throughout life to prevent ovary cells from becoming more like the Sertoli cells present in the testes. Similarly, a gene called *Dmrt1* keeps Sertoli cells from becoming more like ovary cells after birth.

Scientists don't yet know all the details about how *Dmrt1* prevents testes from becoming more ovary-like. For example, do genes that help testes develop in the embryo (which include two genes called *Sox8* and *Sox9*) play a role in maintaining the adult testes?

Barrionuevo, Hurtado, Kim et al. have now genetically engineered adult male mice to lack the *Sox8* and *Sox9* genes. The Sertoli cells in the testes of these mice gradually lost their key characteristics and ultimately died. During this process, the testes cells took on certain characteristics that made them more ovary-like: for example, the ovary-maintaining *Foxl2* gene was activated in the Sertoli cells.

Eventually, the structures in the testes that produce sperm degenerate and are replaced by empty space in the genetically engineered mice. This happens because the *Sox8* and *Sox9* genes control the production of proteins that maintain these structures. In addition, these genes also protect the Sertoli cells from self-destructing, and the testes-maintaining *Dmrt1* gene is not active when *Sox8* and *Sox9* are missing. More studies are now needed to determine how *Sox8* and *Sox9* work with *Dmrt1* to maintain the testes.

signaling pathway becomes active and induces ovarian development (*Sekido and Lovell-Badge, 2008*; reviewed in *Svingen and Koopman, 2013*; *Sekido and Lovell-Badge, 2013*). Both pathways antagonize each other: the loss of either *SRY* or *SOX9* leads to the formation of XY ovaries (*Berta et al., 1990*; *Foster et al., 1994*; *Wagner et al., 1994*) whereas the absence of WNT-signaling molecules such as WNT4 or RSPO1 causes XX sex reversal (*Vainio et al., 1999*; *Parma et al., 2006*). Similarly, gain-of-function experiments confirmed this antagonism, as either upregulation of the testis promoting genes *Sox9* or *Dmrt1* in the XX bipotential gonad (*Bishop et al., 2000*; *Vidal et al., 2001*; *Zhao et al., 2015*) or ectopic activation of the canonical WNT signaling pathway in the XY bipotential gonad (*Maatouk et al., 2008*) leads to XX and XY sex reversal, respectively. Furthermore, Sertoli cell-specific conditional inactivation of *Sox9* on a $Sox8^{-/-}$ background at embryonic day 13.5 (E13.5), two days after the sex determination stage, leads to *Dmrt1* downregulation with upregulation of the ovarian-specific genes *Wnt4, Rspo1* and *Foxl2* (*Barrionuevo et al., 2009*; *Georg et al., 2012*). Similarly, Sertoli cell-specific ablation of *Dmrt1* at the same stage (E13.5) results in ectopic expression of *Foxl2* by postnatal day 14 (P14) and to *Sox9* downregulation by P28, including male-to-female genetic reprogramming, as revealed by mRNA profiling (*Matson et al., 2011a*). Again, gain-of-function experiments confirmed the existence of sexual antagonism after the sex determination period, as conditional stabilization of β-catenin in differentiated embryonic Sertoli cells (E13.5, *Amh-Cre*) resulted in testis cord disruption (*Chang et al., 2008*). The male-vs-female genetic antagonism also persists in the adult ovary. The finding that in adult fertile females granulosa cells transdifferentiate into Sertoli-like cells after *Foxl2* ablation revealed that terminally differentiated female somatic cells require permanent repression of the male-promoting factors to maintain correct identity and function (*Uhlenhaut et al., 2009*). Furthermore, transgenic expression of *Dmrt1* in the adult ovary silenced *Foxl2* and transdifferentiated granulosa cells into Sertoli-like, *Sox9*-expressing cells (*Lindeman et al., 2015*).

Regarding the adult testis, a similar phenomenon appears to occur in fully functional Sertoli cells after *Dmrt1* ablation (*Matson et al., 2011a*). In addition to cells with a Sertoli cell morphology expressing both SOX9 and FOXL2, some cells with typical granulosa cell features were also observed, including the absence of SOX9 and the presence of FOXL2. However, Sertoli-to-granulosa cell transdifferentiation was not unambiguously documented, as the authors used an inducible ubiquitous promoter (*UBC-CreERT2*) for *Dmrt1* ablation in adult Sertoli cells and the possible existence of genetic reprogramming was not investigated as no mRNA profiling was performed in adult mutant testes.

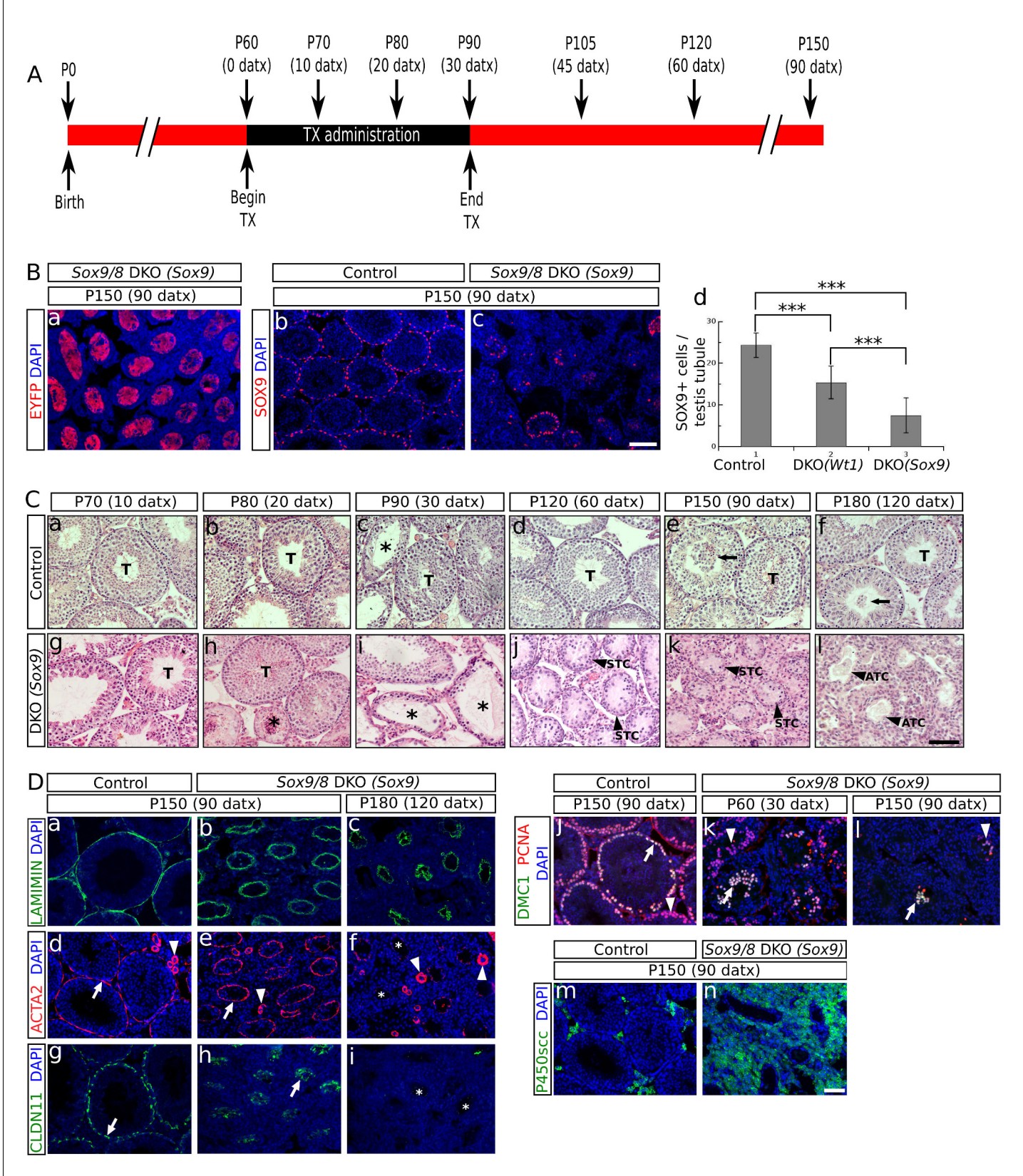

**Figure 1.** *Sox9* and *Sox8* maintain the function and integrity of the adult mouse testis. (**A**) Diagram illustrating the time course of TX administration. Mice were fed with a TX-supplemented diet during one month, between P60 (0 datx) and P90 (30 datx). After this period, mice were fed with a normal

*Figure 1 continued on next page*

*Figure 1 continued*

diet. The main stages studied in this work are depicted. (B) Analysis of the Cre-recombination efficiency in *Sox9/8* DKO (*Sox9*) mice at p150 (90 datx). (a) EYFP is widely expressed in SC-DKO testis cords. At the same stage, the number of SOX9+ cells in the control (*Sox9$^{f/f}$;Sox8$^{-/-}$*) (b) is clearly higher than in the mutant (c). (d) Comparisons of the mean number of SOX9+ cells per transversal testis tubule section in control (*Sox9$^{f/f}$;Sox8$^{-/-}$*) and mutant testes. All pairwise comparisons provided statistically significant differences (two tail test, p<0.001 in all cases). (C) Time-course of testis regression in *Sox9/8* DKO (*Sox9*) mice. Representative micrographs are shown for both TX-treated controls (*Sox9$^{f/f}$;Sox8$^{-/-}$*) (a–f) and *Sox9/8* DKO (*Sox9*) mice (g–l) between p70 (10 datx) and P180 (120 datx). T, normal seminiferous tubules; arrows indicate desquamated germ cells; asterisks mark testis tubules showing signs of degeneration (from enlarged lumen to Sertoli cell-only condition); STC, shrunken testis cords; ATC acellular testis cords. (D) Analysis of somatic (a–i and m–n) and germ cell (j–l) molecular markers. Immunofluorescence for LAMININ (a–c), ACTA2 (d–f) and CLAUDIN11 (g– i) in both P150 (90 datx) TX-treated control (*Sox9$^{f/f}$;Sox8$^{-/-}$*) (a, d and g) and SC-DKO testes at P150 (90 datx) (b, e and h) and P180 (120 datx) (c, f and i). Arrows mark seminiferous tubule expression of ACTA2 (d and e) and Claudin11 (g and h); arrowheads mark arterial expression of ACTA2 (d–f). Asterisks mark testis cords lacking ACTA2 (f) or Claudin11 (i) expression. Double immunofluorescence for PCNA and DMC1 showing the time-course of spermatogenesis reduction in the testes of both P150 (90 datx) TX-treated control (*Sox9$^{f/f}$;Sox8$^{-/-}$*) (j) and P90 (30 datx) and P150 (90 datx) SC-DKO mice (k,l). Arrows mark spermatocytes showing colocalization of the two proteins; arrowheads mark proliferating spermatogonia expressing PCNA but not DMC1. Expression of P450SCC (green fluorescence) in Leydig cells of both TX-treated control (*Sox9$^{f/f}$;Sox8$^{-/-}$*) (m) and SC-DKO (n) testes at P150 (90 datx). Scale bars in Bc, Cl and Dn represent 100 µm for pictures in B, 50 µm for those in C and 50 µm for those in D, respectively.

The following source data and figure supplements are available for figure 1:

**Source data 1.** Comparison of the number of SOX9+ cells per testis tubule in both SC-DKO mutants and TX-treated controls (*Sox9$^{f/f}$;Sox8$^{-/-}$*).

**Figure supplement 1.** Analysis of the CRE-recombination efficiency in SC-DKO mice.

**Figure supplement 2.** Redundant role for *Sox9* and *Sox8* in the maintenance of adult testis cord architecture.

**Figure supplement 3.** Relative abundance of the most relevant morphological features observed in the testes of P120 control and *Sox9/8* mutant mice differing in the number of *Sox9/8* mutant alleles 60 days after TX administration during 5 days with a feeding-gauge needle.

**Figure supplement 4.** Time course of the testis phenotype of control and SC-DKO (*Wt1*) mice.

**Figure supplement 5.** Relative abundance of the most relevant morphological features observed throughout the timecourse of testis regression in double *Sox8/Sox9* mutant mice.

**Figure supplement 6.** Functional status of the BTB in *Sox9/8* DKO testes.

**Figure supplement 7.** Expression bar plots of two adult Leydig cell markers.

Nothing is known on the role of SOX9 in the adult testis, where it is expressed by Sertoli cells in a spermatogenic stage-dependent manner in several mammalian species (*Fröjdman et al., 2000*; *Dadhich et al., 2011*; *Massoud et al., 2014*). Here we report the use of two Sertoli-cell-specific *Cre* lines (*Wt1-CreERT2* and *Sox9-CreERT2*) to induce *Sox9* ablation on a *Sox8$^{-/-}$* background in the adult testis, starting at postnatal day 60 (P60). We show that *Sox9/8* Sertoli cell-specific knockout (SC-DKO) testes undergo testis-to-ovary genetic reprogramming and Sertoli-to-granulosa cell transdifferentiation. The process is retinoic acid (RA)-mediated and occurs as a consequence of *Dmrt1* downregulation. SOX9/8 are necessary to maintain *Dmrt1* expression and thus to prevent *Foxl2* expression in the adult testis. Furthermore, double mutant testes exhibited complete degeneration of the seminiferous tubules and increased apoptosis, indicating that SOX9/8 are continually required for the maintenance of testis integrityy.

## Results

### *Sox9* and *Sox8* are necessary for maintaining the function and integrity of adult seminiferous tubules

To investigate the function of *Sox9* and *Sox8* in the adult testis, we induced the Sertoli cell-specific ablation of *Sox9* in adult *Sox8* null mutant mice using the tamoxifen (TX)-inducible *Cre-loxP*

mutagenesis system. We used two different *CreERT2* mouse lines, a *Wt1* knock-in line (*Wt1-CreERT2*; *Zhou et al, 2008*), and a *Sox9* BAC-transgenic line (*Sox9-CreERT2*; *Kopp et al, 2011*). To check the Cre recombination efficiency, we introduced the R26R-EYFP allele in both *Sox9-CreERT2* and *Wt1-CreERT2* double homozygous *Sox9/8* knockout (DKO) mutants. *Sox8/9* DKO mice fed with a TX-supplemented diet for a maximun of 30 days starting at P60 (*Figure 1A*) showed EYFP expression in a reduced number of Sertoli cells already 10 days after the beginning of TX administration (10 datx, P70) in the two *CreERT2* lines. The number of EYFP⁺ Sertoli cells increased in both lines at later time-points, the *Sox9-CreERT2* line showing always a higher number of positive cells than the *Wt1-CreERT2* line. From P150 (90 datx) on, the EYFP signal occupied the whole area of the seminiferous tubule section (*Figure 1Ba*, *Figure 1—figure supplement 1A*). However, the fact that the cytoplasm of Sertoli cells is very large and complex in shape, together with the severe shrinkage that *Sox9/8* SC-DKO seminiferous tubules have undergone by this time, made it very difficult to identify individual EYFP⁺ cells in these testes. Hence we performed immunofluorescence for SOX9 and counted the number of SOX9⁺ cells per transversal testis cord section. At P90 (30 datx) all seminiferous tubules still contained many positive cells, but the number was clearly reduced by P120 (60 datx) and even more by P150 (90 datx), when some testis cords were completely devoid of SOX9⁺cells (*Figure 1Bb–c*, *Figure 1 —figure supplement 1B*). At this later stage, the number of SOX9⁺ cells per seminiferous tubule cross section decreased to 15.39 ± 3.36 (37% reduction) in the testes of the *Wt1-CreERT2; Sox9^{f/f}; Sox8^{-/-}* [*Sox9/8* DKO (*Wt1*)] mice and to 7.49 ± 3.61 (69% reduction) in those of the *Sox9-CreERT2; Sox9^{f/f}; Sox8^{-/-}* [*Sox9/8* DKO (*Sox9*)] mice, when compared to controls (24.31 ± 2.94) (*Figure 1Bd*, *Figure 1—source data 1*). The fact that the number of recombinant Sertoli cells lacking *Sox9* in these mutant mice continues decreasing for several weeks after the end of the period of TX administration (30 days) suggests that many newly recombined cells appear after that time (persistence of TX in the body) and that perhaps either the *Sox9* transcript or the protein, or both, are very stable in adult Sertoli cells, so that the gene product may remain for days or weeks in the cell after the gene ablation event. We also found that the reduction of SOX9⁺ cells varied among testis cords and among animals. We selected the most affected regions of the most affected individuals for further analyses.

Consistent with the situation reported for embryonic stages of development (*Barrionuevo et al., 2009*), we observed that the testis phenotype of the different *Sox9/8* compound mutants increased in severity with the number of *Sox9/8* mutant alleles (*Figure 1—figure supplement 2* and *3*).

*O'Bryan et al. (2008)* reported a *Sox8^{-/-}* mouse line in which a progressive deregulation of spermatogenesis occurred and where male mice became sterile by P150. In contrast, our *Sox8* mutants (*Sock et al., 2001*) do not show such a severe testicular phenotype and males are normally fertile even at P180. At the histological level, our *Sox8^{-/-}* mice appeared normal until P120, but showed signs of germ cell desquamation (sloughing) afterwards (*Figure 1—figure supplement 4a–f*). Genetic background differences between the two *Sox8^{-/-}* lines may explain these phenotypic discrepancies. TX-treated controls were similar to untreated males, except between P80 (20 datx) and P120 (60 datx) and mainly at P90 (30 datx), when they showed some degenerating seminiferous tubules, but recovered afterwards (*Figure 1Ca–f*, *Figure 1—figure supplement 4a–l*). Testes in *Sox9/8* DKO (*Sox9*) mice were similar to the TX-treated controls at P70 (10 datx) except for a few testis tubules with enlarged lumen (*Figure 1Cg*). At P80 (20datx), only few seminiferous tubules showed signs of degeneration (shrinkage and germ cell depletion), whereas this was more frequent by P90 (30 datx). In many cases, Sertoli cell-only tubules were visible (*Figure 1Ch,i*). By P120 (60 datx), tubules had become solid testis cords whose diameter appeared even more reduced at P150 (90 datx) (*Figure 1Cj,k*). While some mice continued to exhibit this phenotype at P180 (120 datx), a subset of mice in this group was more affected. In these latter mice Sertoli and germ cells had disappeared completely (*Figure 1Cl*). At later time points, all mice showed this severe testicular phenotype. This progressive degeneration of the testicular phenotype in *Sox9/8* SC-DKO mice was evident when we analyzed the relative abundance of the most relevant testicular morphological features between P70 (10 datx) and P180 (120 datx) (*Figure 1—figure supplement 5*). In contrast, Leydig cells appeared morphologically normal in mutant testes. *Sox9/8* DKO (*Wt1*) mice exhibited a similar testicular phenotype (*Figure 1—figure supplement 4m–r*). These results show that *Sox8* and *Sox9* alleles act redundantly in adult Sertoli cells and are necessary to maintain the integrity of the seminiferous tubules of functional testes.

## Functional status of both somatic and germ cells in testes with *Sox9/8*-deficient Sertoli cells

To better define the mutant phenotype, we next studied the expression of several somatic and germ cell markers. Laminin, a principal component of the basement membrane (*Richardson et al., 1995*) persisted in both P150 (90 datx) and P180 (120 datx) testes of SC-DKO mice (*Figure 1Db,c*). *Alpha smooth muscle actin* (*Acta2*) expressed by both peritubular myoid (PM) cells and arterialmuscle fibers was detected in the testes of both TX-treated controls and P150 (90 datx) SC-DKO mice (*Figure 1Dd, e*). In contrast, at P180 (120 datx), strong arterial ACTA2 signal persisted but that of PM cells was almost undetectable (*Figure 1Df*). This shows that acellular cords in severely affected SC-DKO testes have lost not only Sertoli and germ cells, but also PM cells. Claudin11 is a principal component of tight junctions, the main junctional structures forming the blood-testis barrier (BTB). *Cldn11* (the claudin11 gene) expression was similar between controls and double mutants before P150 (90 datx) (not shown), but it was severely reduced by P150 (90 datx) and completely absent in P180 (120 datx) *Sox9/8* mutant testes (*Figure 1Dg–i*), indicating that the BTB is not functional in these testes. To proof this assumption, we tested the permeability of the BTB of P120 (60 datx) mice with a biotin tracer experiment revealing that control testes had a functional BTB, whereas that of the mutant testes had become permeable (*Figure 1—figure supplement 6*). We also performed immunofluorescence for both PCNA, which is expressed in mitotic spermatogonia as well as in zygotene and early pachytene, but not leptotene spermatocytes (*Chapman and Wolgemuth, 1994*), and DMC1, a meiotic recombination protein marking zygotene-pachytene spermatocytes (*Yoshida et al., 1998*). At P60 (30 datx), most mutant seminiferous tubules exhibited a clear reduction of spermatogenic activity and some spermatocytes were abnormally located in the inner region of the tubules (*Figure 1Dk*) and not at the periphery, as seen in TX-treated control testes (*Figure 1Dj*). In P120 (60 datx) testes, spermatocytes were scarce and only proliferating spermatogonia were seen in most testis tubules (not shown), while at P150 (90 datx), both spermatogonia and spermatocytes had disappeared in most tubules (*Figure 1Dl*). These results indicate that spermatogenesis becomes disrupted in testes with Sertoli cells deficient for both *Sox9* and *Sox8*. Unlike other somatic cells, Leydig cells appear not to be seriously affected in testes from *Sox9/8* SC-DKO mice. These cells do not transdifferentiate into theca cells, as they never express *Foxl2* (as theca cells do; not shown), and maintain the steroidogenic function for a long time after *Sox9* ablation, as deduced from the expression of P450scc, a cytochrome involved in the synthesis of testosterone (*Figure 1Dm,n*). Consistently, the testosterone-producing enzyme *HSD17b3* and the marker for adult functional Leydig cells *Insl3* are expressed at high levels in the mutant testes (*Figure 1—figure supplement 7*).

## Somatic testis-to-ovary genetic reprogramming in the absence of *Sox9* and *Sox8* in adult mouse testes

The loss of *Foxl2* in adult granulosa cells results in a somatic ovary-to-testis genetic reprogramming with granulosa-to-Sertoli cell transdifferentiation which includes *Sox9* upregulation (*Uhlenhaut et al., 2009*). Contrarily, *Foxl2* is upregulated when *Sox9* is ablated in embryonic Sertoli cells of *Sox8* null mutants after the sex-determination stage (*Georg et al., 2012*). To test whether a similar phenomenon took place in our *Sox9/8* SC-DKO mice, we carried out immunofluorescence for FOXL2. At P90 (30 datx), FOXL2 protein was almost completely absent from mutant testes. However, by P105 (45 datx), positive cells were present in almost all testis cords, and by P150 (90 datx), the most severely affected mice showed many FOXL2-positive cells within almost all testis cords (*Figure 2A*, *Figure 2—figure supplement 1*). These results show that transdifferentiation also occurs in adult *Sox9/8* DKO Sertoli cells. Accordingly, we performed a genome-wide transcriptome analysis of P150 untreated control testis, P150 (90 datx) control and mutant testis and control ovary. Our results show that SC-DKO testes exhibit a striking feminization of the testicular transcriptome. *Figure 2B* shows a $Log_2$-fold-change heat map including the 12,380 genes detected to have significant differential expression between the five sample conditions (the complete list of genes with differential expression is shown in *Figure 2—source data 1*). With the exception of a few gene clusters, most genes in mutant testes adopted an ovary-like expression pattern (*Figure 2B*, *Figure 2—figure supplement 2*, *Supplementary file 1*). Cluster analyses of all genes, both by replicates and by conditions, showed that mutants are clustered together, with no clear distinction between *Sox9-CreER* and *Wt1-CreER* lines (*Figure 2—figure supplement 3*). Similarly, pairwise gene sets with significant differential

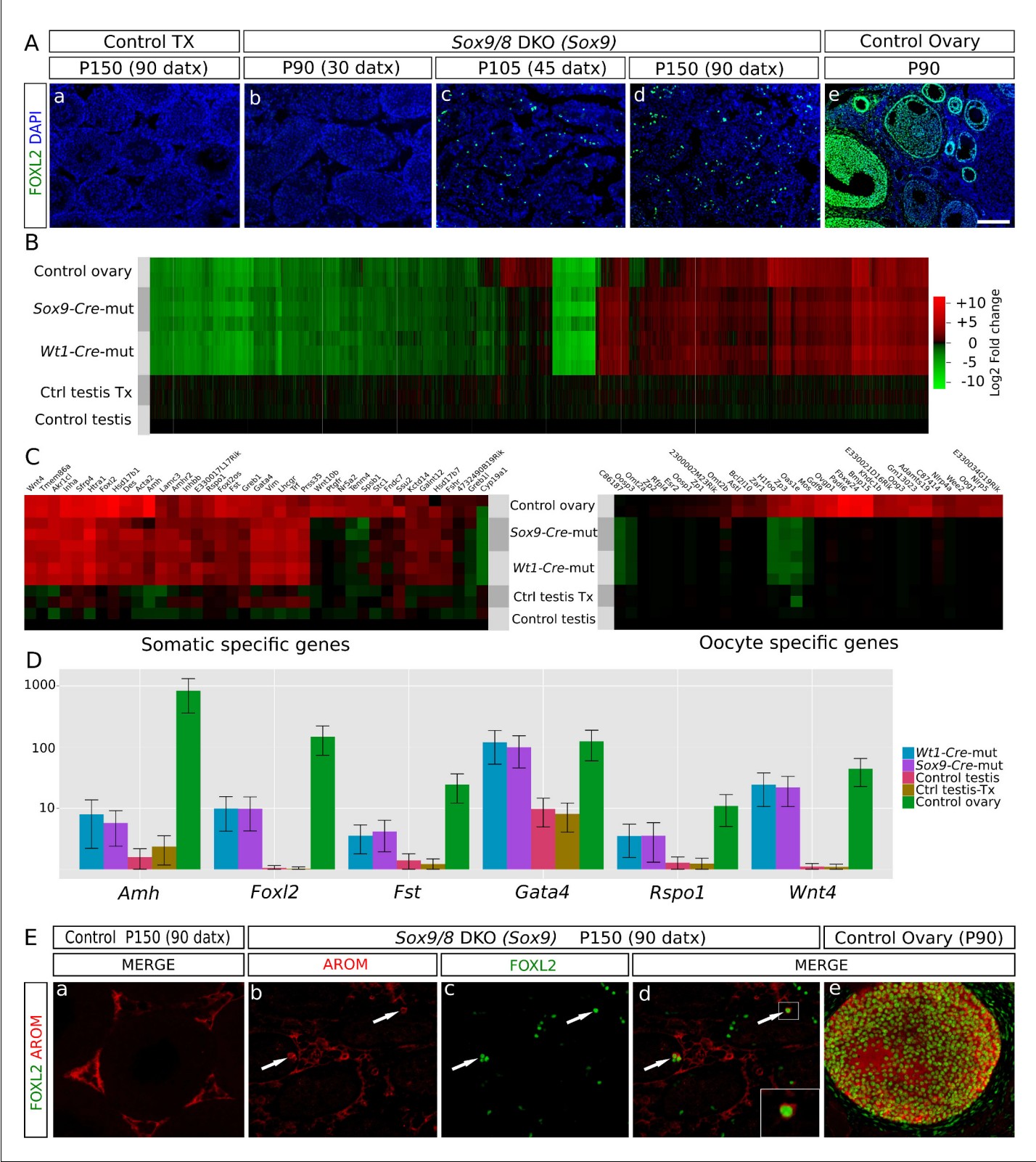

**Figure 2.** Genetic reprogramming in somatic cells of adult *Sox9/8* SC-DKO (*Sox9*). (**A**) Expression of FOXL2 (green fluorescence) in P150 (90 datx) TX-treated control (*Sox9^{f/f};Sox8^{-/-}*) (a) and in *Sox9/8* SC-DKO (*Sox9*) mouse testes analyzed at P90 (30 datx) (b), P105 (45 datx) (c), and P150 (90 datx) (d) as well as in a P 90 control ovary (e). (**B**) Heatmap showing the 12,380 genes found to be differentially expressed at alpha < 0.005 when comparing control (*Sox9^{f/f}*) and mutant adult gonads. The log_2(FPKM+1) of each gene in each condition has been divided by the corresponding value in control testis. *Figure 2 continued on next page*

*Figure 2 continued*

Gene expression has not been altered by the TX treatment. Red colors indicate genes upregulated with respect to their expression levels in control testis and green colors indicate downregulated genes. (**C**) Expression heatmaps of selected ovarian somatic-specific and oocyte-specific genes. (**D**) Expression bar plots of six relevant ovarian somatic-specific genes upregulated in mutant testes. (**E**) Aromatase (red) and FOXL2 (green) immunofluorescence staining of TX-treated control (*Sox9^{f/f};Sox8^{-/-}*) testis (a), mutant testes (b–d), and control ovary (e). Arrows mark reprogrammed Sertoli cells showing simultaneous expression of Aromatase and FOXL2. Scale bar shown in Ae represents 150 μm in A and 75 μm in E.

The following source data and figure supplements are available for figure 2:

**Source data 1.** Genes with significant differential expression among untreated controls, TX-treated controls, *Sox8/9* SC-DKO mutants and control ovary at P150 (90 datx) identified from the bioinformatic analysis of our transcriptome.
**Figure supplement 1.** Expression of *Foxl2* in somatic cells of adult *Sox9/8* DKO (*Wt1*).
**Figure supplement 2.** Heatmaps showing the expression of genes involved in 8 selected pathways (A–H), relative to their expression in control testes.
**Figure supplement 3.** Cluster analysis of (A) replicates and (B) conditions.
**Figure supplement 4.** Quantification of genes with differential expression and Jensen–Shannon (JS) distances between conditions.

expression at $\alpha < 0.05$ demonstrated that the number of differentially expressed genes is higher when mutants were compared with testis controls than when compared with ovary (*Figure 2—figure supplement 4A*). Accordingly, the distance map is higher between mutant and control testis than between mutant testis and ovary (*Figure 2—figure supplement 4B*). The same results were obtained when comparing isoforms, transcription start sites or coding DNA sequences (not shown). Expression heat maps for selected 39 ovarian somatic cell-specific genes and 33 oocyte-specific genes selected using bioGPS (biogps.gnf.org) revealed that the cell reprogramming observed in the SC-DKO testes only affects somatic cells (*Figure 2C*). Notably, bar plots for six genes known to be adult granulosa cell markers showed that these genes were upregulated in the mutant Sertoli cells, revealing an ovary-like expression pattern (*Figure 2D*). In addition, within the seminiferous cords of SC-DKO testes we found a few FOXL2$^+$ cells expressing the enzyme aromatase (*Figure 2E*). This is evidence that, in addition to *Foxl2*, other genes normally expressed by granulosa cells are also transcribed and translated in *Sox9/8* SC-DKO testes.

## Sertoli-to-granulosa cell transdifferentiation in adult *Sox9/8* SC-DKO testes

We next investigated the origin of the granulosa-like, FOXL2$^+$ cells present in the *Sox9/8* SC-DKO testes. Several pieces of evidence show that FOXL2$^+$ cells in our mutant testes originate from *Sox9/8* null Sertoli cells. The two gene promoters we used to drive *Cre* expression (*Sox9* and *Wt1*) are Sertoli cell-specific in the testis, indicating that transdifferentiation originates directly from this cell type. Importantly, we found that FOXL2$^+$ cells always located inside testis cords with strong expression of the Cre-recombination reporter EYFP (*Figure 3A*). We also analyzed the expression of WT1, a Sertoli cell marker whose expression is maintained after *Sox9/8* ablation in embryonic mouse Sertoli cells (*Barrionuevo et al., 2009*), and that it is co-expressed with FOXL2 in granulosa cells of immature, but not mature, follicles (*Chun et al., 1999*; *Figure 3Ce*). At P90 (30 datx) we observed many WT1$^+$ Sertoli cells that have already lost SOX9 (green cells, *Figure 3Bb*). The number of WT1$^+$ SOX9$^-$ cells decreased by P150 (90 datx) (*Figure 3Bc*), indicating that recombined Sertoli cells were being lost. This decrease in the number of WT1$^+$ SOX9$^-$ Sertoli cells coincides with an increase in the number of FOXL2$^+$ cells which either retain weak WT1-staining or are WT1$^-$ (*Figure 3C*), suggesting that FOXL2$^+$ cells originate from cells previously expressing WT1, that is Sertoli cells. Altogether, these results indicate that *Sox9/8* SC-DKO testes experience a cell-autonomous Sertoli-to-granulosa cell transdifferentiation which triggers the observed testis-to-ovary genetic reprogramming in these gonads.

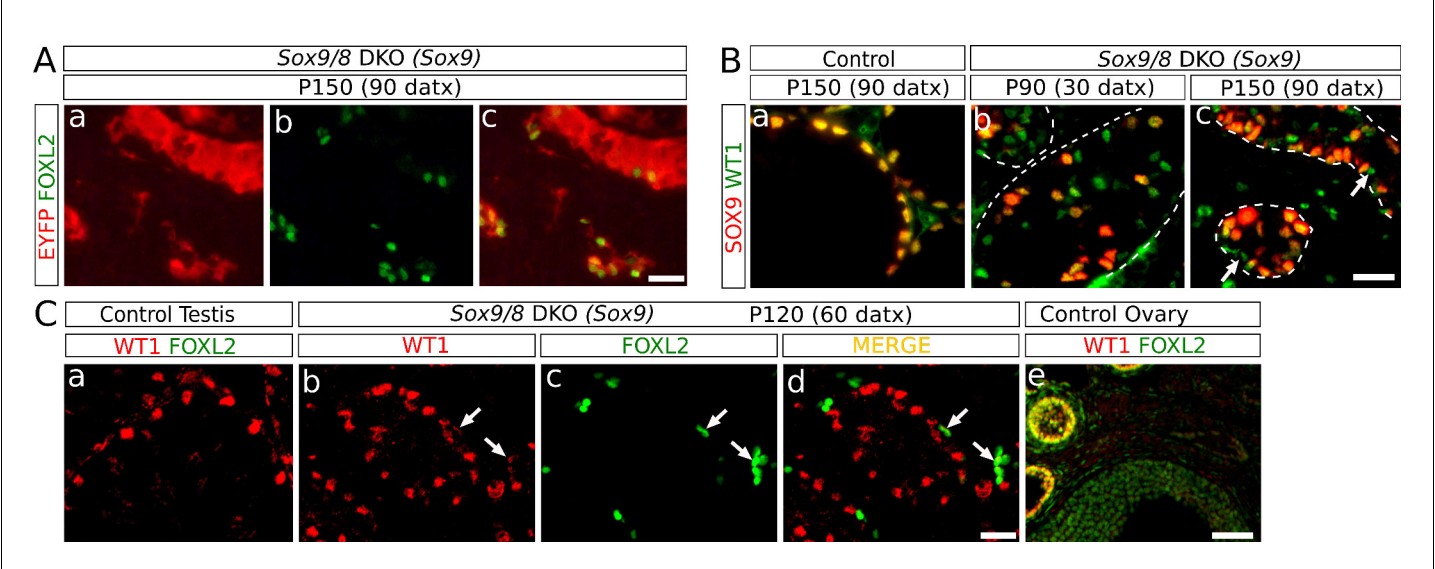

**Figure 3.** Identification of the somatic cells expressing FOXL2. (A) Double immunofluorescence for FOXL2 (green, nuclear) and EYFP (red, cytoplasmic) (B) Double immunofluorescence for SOX9 and WT1 in P150 (90 datx) TX-treated control ($Sox9^{f/f};Sox8^{-/-}$) testes (a) and $Sox9/8$ SC-DKO testes analyzed at P90 (30 datx) (b) and P150 (90 datx) (c). Dashed lines delineate the seminiferous tubules contour. Arrows mark mutant cells expressing WT1 but not SOX9. (C) Double immunofluorescence for FOXL2 and WT1 in both TX-treated control ($Sox9^{f/f};Sox8^{-/-}$) (a) and $Sox9/8$ SC-DKO mutant testes analyzed at P120 (60 datx) (b–d), as well as in a control ovary (e). Arrows point to mutant cells expressing both proteins. Scale bar in **Ac** represents 25 μm in **A**; scale bar in **Bc** represents 25 μm in **B**; scale bar in **Cd** represents 25 μm in **Ca–d** and scale bar in **Ce** represents 100 μm.

## Sertoli-to-granulosa cell transdifferentiation is mediated by *Dmrt1* downregulation in *Sox9/8* SC-DKO testes

Since *Sox9* is upregulated after *Foxl2* ablation in adult granulosa cells (*Uhlenhaut et al., 2009*) and downregulated after *Dmrt1* ablation in embryonic Sertoli cells (*Matson et al., 2011a*), we investigated the expression pattern of these two genes in the testes of the *Sox9/8* SC-DKO mutants. We found that cells coexpressing SOX9 and FOXL2 were rare at any stage analyzed [12 out of 203 FOXL2+ cells co-expressed SOX9 at P120 (60 datx)] (*Figure 4A*, *Figure 4—figure supplement 1A*), indicating that *Foxl2* upregulation requires previous elimination of both SOXE proteins. Next, we examined the expression of both *Sox9* and *Dmrt1* in *Sox9/8* SC-DKO testes. As *Dmrt1* is expressed in both Sertoli cells and spermatogonia of adult testes (*Raymond et al., 2000*), we used a third marker, PCNA, that labels spermatogonia as well as zygotene and early pachytene spermatocytes. Whereas all Sertoli cells in control testes showed strong staining for both DMRT1 and SOX9 (SS, *Figure 4Ba*), mutant Sertoli cells showed varying degrees of both SOX9 and DMRT1 staining intensity, although they normally paralleled each other in intensity. Therefore Sertoli cells with a weak staining for both DMRT1 and SOX9 (WS) were also visible in these testes. Consistent with this, we found a very reduced number of cells expressing only DMRT1 at P90 (30 datx) (*Figure 4Bb–d*, red cells (arrow); SOX9- DMRT1+ PCNA-) and almost none at P120 (60 datx) (*Figure 4Be–g*). Furthermore, in P150 (90 datx) testes, which are almost devoid of germ cells, DMRT1 immunoreactivity was almost exclusively restricted to SOX9+ cells (*Figure 4—figure supplement 1B*). Double WT1-DMRT1 staining confirmed that as early as at P90 (30 datx) many WT1+ cells (Sertoli cells) have already lost DMRT1 expression (green cells in *Figure 4Cb*), showing that *Dmrt1* is downregulated after *Sox9* ablation and before *Wt1* downregulation occurs in SC-DKO testes. In addition, as observed for SOX9 and FOXL2 (see above), DMRT1 and FOXL2 only colocalize in a reduced number of cells in the testes of our *Sox9/8* SC-DKO mice [16 out of 127 FOXL2+ cells co-expressed DMRT1 at P120 (60 datx)] (*Figure 4D*, *Figure 4—figure supplement 1C*). Overall, these findings support the notion that SOX9 and SOX8 are necessary for the maintenance of *Dmrt1* expression in adult Sertoli cells and that these testis-promoting factors negatively regulate *Foxl2*.

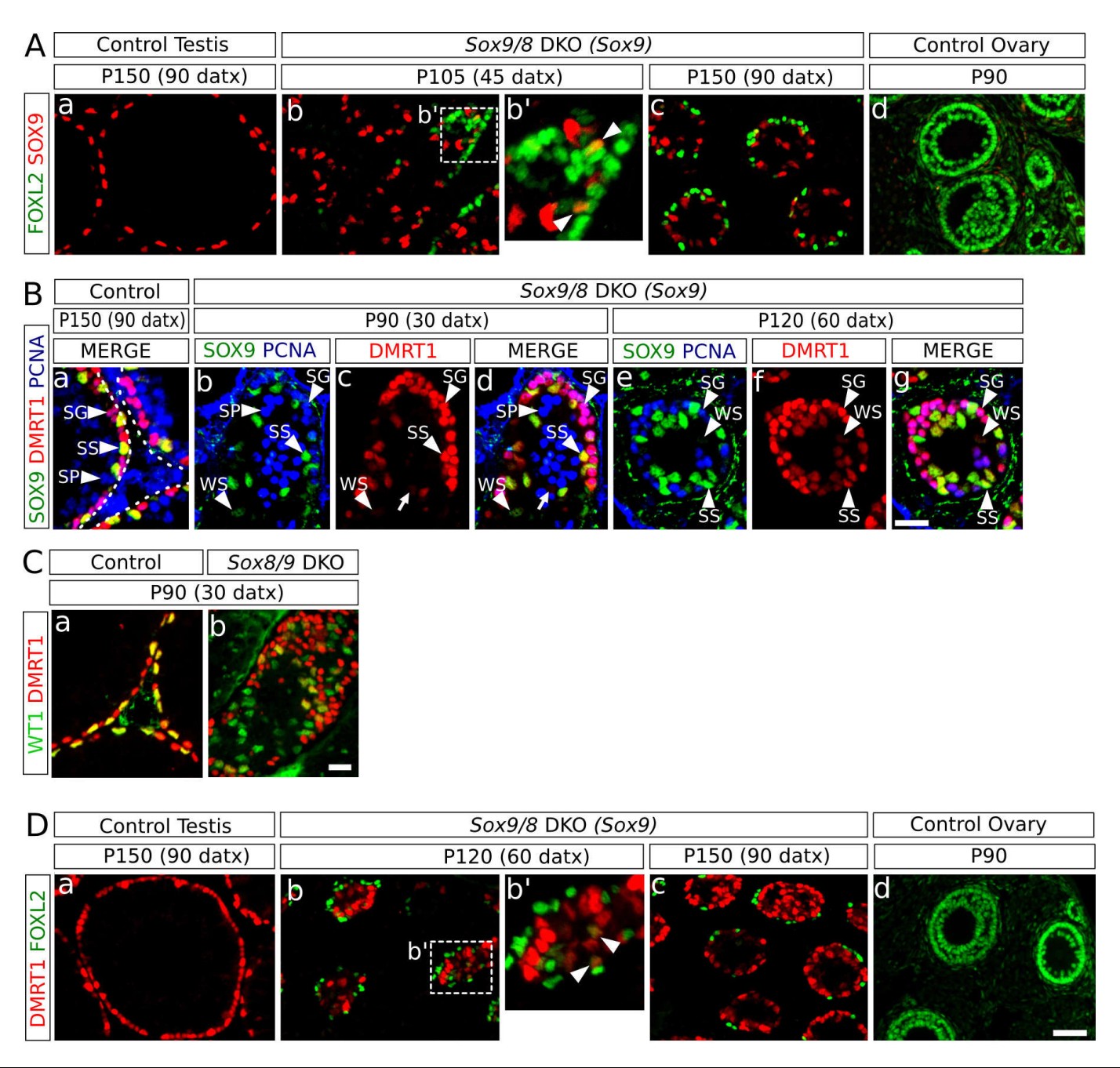

**Figure 4.** Role of *Dmrt1* in Sertoli-to-granulosa cell transdifferentiation. (**A**) Double immunofluorescence for SOX9 and FOXL2 in TX-treated control (*Sox9^{f/f};Sox8^{-/-}*) testis (a), *Sox9/8* SC-DKO mutant testes analyzed at P105 (45 datx) and P150 (90 datx) (b,c) and a control ovary (d) (b' is a higher magnification of the area marked in b). Colocalization of SOX9 and FOXL2 was rare and the few observed cells showed weak fluorescence for both proteins (arrowheads in b'). (**B**) Triple immunofluorescence for SOX9, DMRT1 and PCNA (germ cell marker) in P150 (90 datx) TX-treated control (*Sox9^{f/f}; Sox8^{-/-}*) testes (a) and *Sox9/8* SC-DKO mutant testes at P90 (30 datx) (b–d) and P120 (60 datx) (e-g). Different cell types can be identified: SS: Sertoli cells with strong staining for both DMRT1 and SOX9 (SOX9^+ DMRT1^+ PCNA^-; strong yellow); WS: Sertoli cells with weak staining for both DMRT1 and SOX9 (SOX9^+ DMRT1^+ PCNA^-; pale yellow); SP: spermatocytes (SOX9^- DMRT1^- PCNA^+; blue); SG: proliferating spermatogonia (SOX9^- DMRT1^+ PCNA^+; purple), arrow (SOX9^- DMRT1^+ PCNA^-; red). Non-proliferating spermatogonia could be confused in *Sox9/8* SC-DKO mice with recombined DMRT1^+ SOX9^- Sertoli cells in which SOX9 already disappeared, but the number of the former cell type is so low that they can be ignored. (**C**) Double immunofluorescence for DMRT1 and WT1 in P90 (30 datx) TX-treated control (*Sox9^{f/f};Sox8^{-/-}*) (a) and mutant testes (b). (**D**) Double immunofluorescence for DMRT1 and FOXL2 in P150 (90 datx) TX-treated control (*Sox9^{f/f};Sox8^{-/-}*) testis (a), *Sox9/8* SC-DKO mutant testes (b–c) and control ovary (d) (b' is a

*Figure 4 continued on next page*

*Figure 4 continued*

higher magnification of the area marked in b). Colocalization of both proteins was rare (arrowheads in b'). Scale bar in **Dd** represent 50 µm in **A** and **D**; scale bar in **Bg** represents 25 µm in **B**; scale bar in **Cb** represents 50 µm in **C**.

The following figure supplement is available for figure 4:

**Figure supplement 1.** Role of Dmrt1 in Sertoli-to-granulosa cell transdifferentiation.

To further test this hypothesis, we compared the microarray data from P28 SC-*Dmrt1* KO testes reported by *Matson et al. (2011b)* with the RNA-seq data from our P150 (90 datx) SC-*Sox9/8* DKO testes, and plotted all mRNAs that resulted either downregulated or upregulated when compared to control males in both datasets (*Figure 5A*, small blue dots). Nearly all genes strongly affected by the loss of *Dmrt1* were also affected by the loss of *Sox9/8* (*Figure 5A*, *Figure 5—source data 1*). This

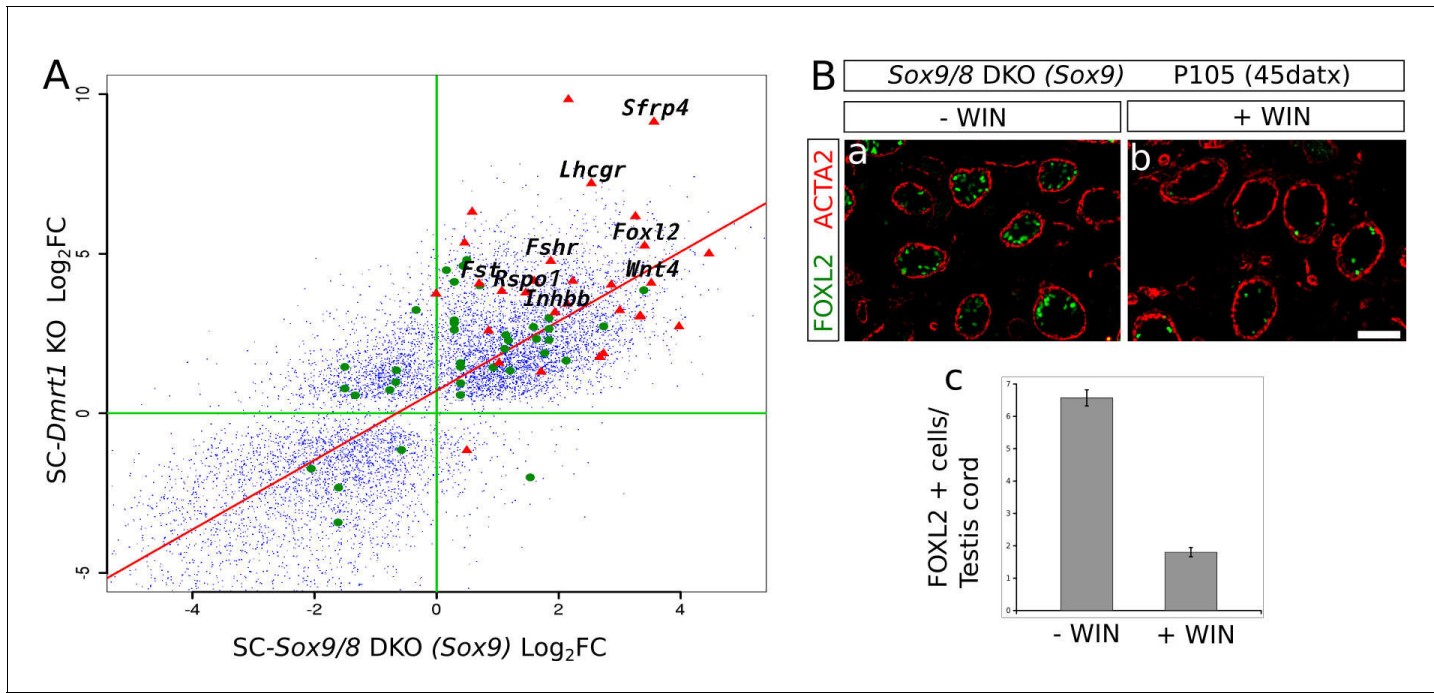

**Figure 5.** *Sox9* and *Dmrt1* act in the same pathway during Sertoli-to-granulosa transdifferentiation. (**A**) Log$_2$ fold change scatterplot comparing the microarray data from the P28 SC-*Dmrt1* KO testes reported by *Matson et al., 2011b*; GEO accession: GSE27261) with the RNA-seq data from our P150 (90 datx) *SC-Sox9/8* DKO testes, including 8910 genes showing significant differential expression respect to normal testis in both *Dmrt1* and *Sox9/8* mutants (blue dots). Among these, green dots represent 24 genes belonging to the all-trans-retinoic acid-mediated apoptosis path and RA receptors-mediated signaling from the PathCards database. Red triangle show 29 of the ovarian somatic specific genes included in *Figure 2C*. The names of some relevant genes are indicated. Regression line is shown in red (intecept = 0.7112, slope = 1.0883). (**B**) Effect of the treatment of *Sox9/8* SC-DKO mice with WIN 18,644 on Sertoli-to-granulosa cell transdifferentiation. FOXL2-positive cells (green fluorescence) were much more abundant in untreated (**a**) than in treated mutant testes (**b**). The number of positive cells per testis/cord section was 3.5-fold higher in untreated mice (**c**). The ACTA2 signal (red fluorescence) permitted to delineate the testis cords in a and b. Scale bar in **Bb** represents 100 µm in **Ba–b**.

The following source data is available for figure 5:

**Source data 1.** List of 8910 genes showing significant differential expression respect to normal testis in both *Dmrt1* and *Sox9/8* mutants.

**Source data 2.** Log$_2$ fold change of expression of both *Dmrt1* and *Sox9/8* mutants respect to controls in a set of genes belonging to the all-trans-retinoic acid-mediated apoptosis path and RA receptors-mediated signaling from the PathCards database.

**Source data 3.** Comparison of the number of FOXL2[+] cells per transversal testis cord section in *Sox9/8* DKO (*Sox9*) WIN 18,446-treated mice and vehicle-injected controls

finding suggests that both *Dmrt1* and *Sox9/8* act in the same pathway, although the possibility also exists that this coincidence between both gene expression patterns could be a secondary effect of the change in relative numbers of cell types in the SC-*Sox9/8* DKO testes. Among the genes upregulated in both experiments (upper right quadrant in *Figure 5A*), we found 29 somatic ovarian-specific genes including female promoting genes such as *Foxl2, Wnt4, Rspo1, Fst, Fshr* (*Figure 5A*, red triangles). Also, a set of genes were upregulated in *Dmrt1* mutants and downregulated in *Sox9/8* mutants (upper-left quadrant in *Figure 5A*), which may be a consequence of 1) the age-differences between the two compared sample sets, 2) the incomplete efficiency of *Sox9* inactivation of our conditional SC-*Sox9* KO, or 3) the existence of additional roles for *Sox9/8* and/or *Dmrt1* in the adult testis.

It was recently reported (*Minkina et al., 2014*) that DMRT1 functions by protecting male gonadal cells from retinoid acid (RA)-dependent sexual transdifferentiation and that this process could be inhibited by blocking intra-tubular RA synthesis in the *Dmrt1*-mutant testes. By comparing the mRNA profiling of SC-*Dmrt1* KO and *SC-Sox9/8* DKO testes, we found a set of genes belonging to the RA-signaling pathway showing similar misexpression in both mutants (*Figure 5A*, green dots, *Figure 5—source data 2*). As *Dmrt1* is downregulated in the *Sox9/8* SC-DKO testes, we hypothesized that reducing RA levels in our *SoxE* mutants should also affect the transdifferentiation process. To test this, we treated adult SC-DKO mice with the retinaldehyde dehydrogenase inhibitor WIN 18,446 just when the first FOXL2-positive cells are detected. We found a 3.5-fold reduction in the number of FOXL2-positive cells per testis cord section in the WIN 18,446-treated mice (1.80 ± 2.03), compared to the vehicle (DMSO)-injected controls (6.57 ± 3.52; p<0.001; *Figure 5B*, *Figure 5—*

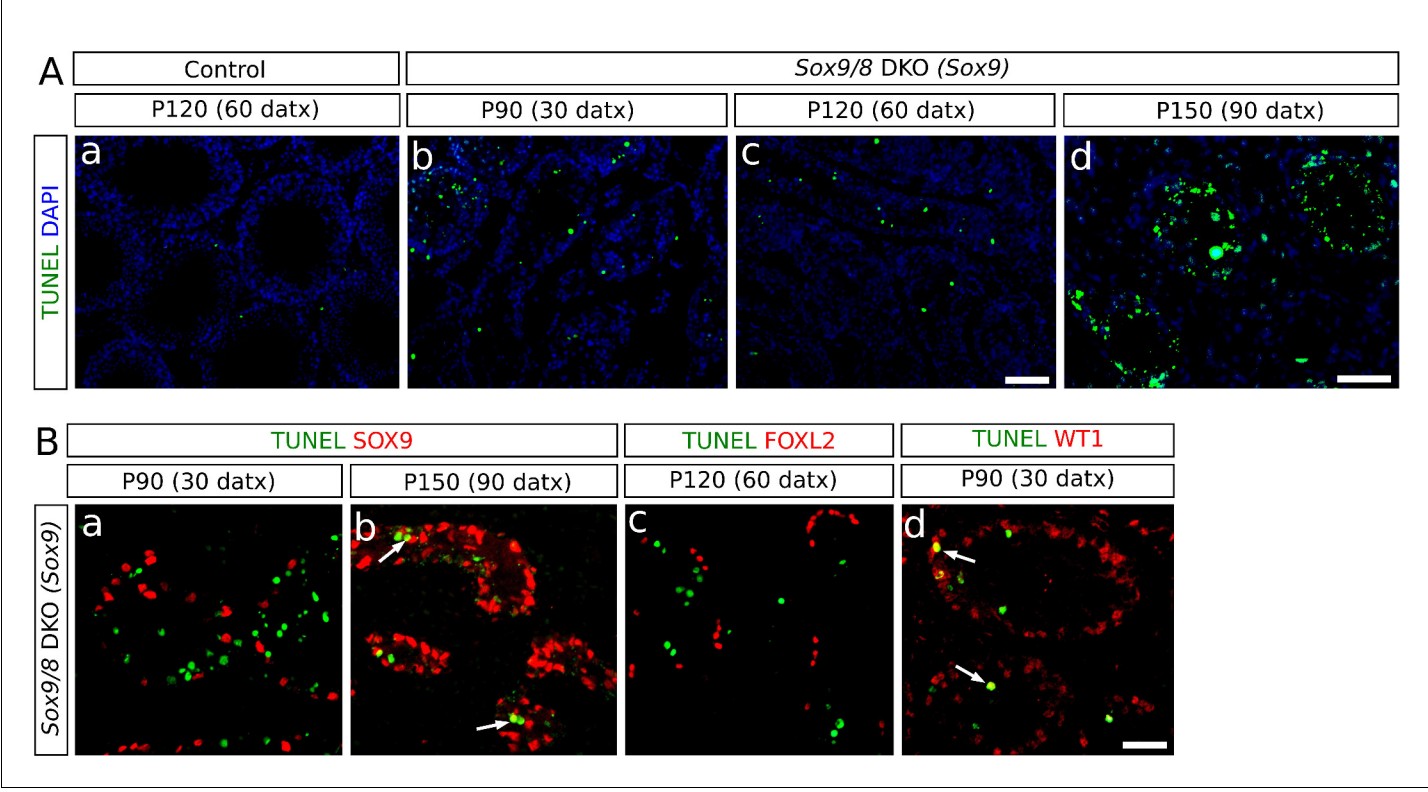

**Figure 6.** Incidence of apoptosis in *Sox9/8* SC-DKO testes. (**A**) TUNEL staining in testis sections of TX-treated control (*Sox9f/f;Sox8−/−*) at P120 (60 datx) (a) and *Sox9/8* SC-DKO at different time-points (b–d). (**B**) Double immunofluorescence for TUNEL and three molecular markers: SOX9 [a, (P90 (30datx)) and b, (P150 (90 datx))], FOXL2 (c, P120 (60 datx)), and WT1 (d, P90 (30 datx)). Arrows in b and d mark cells showing coexpression of the two proteins. Scale bar in Ac represents 100 μm for **Aa–c**; scale bar in **Ad** represents 50 μm; scale bar in **Bd** represents 50 μm in **B**.

The following source data is available for figure 6:

**Source data 1.** Comparison of the number of TUNEL-positive cells per section area unit in both SC-DKO mutants and TX-treated controls.

*source data 3*). Hence, as reported for *Dmrt1* SC-KO mice (*Minkina et al., 2014*), the process of Sertoli-to-granulosa cell transdifferentiation seems to be also inhibited when RA levels were reduced in our study model.

## SOX9 and SOX8 act as anti-apoptotic factors in adult Sertoli cells

Coinciding roughly with the end of TX treatment, *Sox9/8* SC-DKO testes begin to progressively degenerate, as evidenced by shrinkage of the seminiferous tubules, which in the most severely affected mice reach an extreme degree of tubular involution and become acellular testis cords. A possible explanation for the loss of tubular somatic cells is that apoptosis is operating in these testes. TUNEL assay revealed apoptotic cells mainly inside the testis tubules/cords, showing that interstitial cells (mostly Leydig cells) are not seriously affected. The numbers of TUNEL-positive cells counted in a total area of 11.55 mm$^2$ between P90 (30 datx) and P120 (60 datx) in both SC-DKO mutants (370 cells for the *Wt1-CreERT2* line and 488 cells for the *Sox9-CreERT2* line) were significantly higher than those found in TX-treated control testes (120 cells; goodness of fit test p<2.2e-16 in both cases; *Figure 6Aa–c*, *Figure 6—source data 1*). The presence of abundant apoptotic bodies at P150 (90 datx) (*Figure 6Ad*) documents the massive cell death that had occurred during previous stages in the *Sox9/8* SC-DKO mice.

To identify the cell types undergoing apoptosis, we combined TUNEL staining with immunofluorescence for several molecular markers. Neither SOX9- nor FOXL2-expressing cells were observed to be apoptotic in mutant testes before P120 (60 datx) (*Figure 6Ba,c*), but SOX9$^+$ cells were found to be apoptotic in the P150 (90 datx) testes (*Figure 6Bb*). In contrast, we observed apoptotic cells expressing WT1 as early as P90 (30 datx) (*Figure 6Bd*), indicating that apoptosis mainly affects recombined Sertoli cells in which *Sox9* had been ablated but *Foxl2* had not yet been upregulated. Altogether, these findings suggest that testis regression in *Sox9/8* mutants occurs in two different stages. During the first two months after the initiation of TX administration, both non-recombined Sertoli cells (SOX9$^+$) and transdifferentiated cells (FOXL2$^+$) remain alive, whereas recombined but not yet transdifferentiated cells (SOX9$^-$, WT1$^+$) do undergo apoptosis. In the second stage (P180 and older mice), massive apoptosis affects all cell types, including the remaining Sertoli cells and granulosa-like cells.

## Discussion

### The battle of sexes persists beyond the sex determination stage

There is now compelling evidence that the bipotential nature of the genital ridge at the beginning of gonad development is not completely lost once either testes or ovaries acquire their final adult morphology and functionality. During embryonic development the newly formed Sertoli cells can transdifferentiate to their ovarian counterparts when the testis promoting factors *Sox9* or *Dmrt1* are lost (*Georg et al., 2012*; *Matson et al., 2011a*). The finding that *Foxl2* in the adult ovary was necessary to prevent granulosa-to-Sertoli cell transdifferentiation revealed that this antagonism also operates in the adult gonad. In the adult testis, the same antagonism also appears to exist, as FOXL2$^+$ cells were observed when *Dmrt1* was ubiquitously deleted (*Matson et al., 2011a*). Here we show that Sertoli-to-granulosa cell transdifferentiation can be induced as well in the adult mouse testis by just deleting two *SoxE* genes, *Sox9 and Sox8.* These results evidence that *Sox9* has a crucial role, not only during sex determination and testis differentiation, but also in adult testis maintenance, where, together with *Sox8* and coordinately with *Dmrt1,* it prevents male-to-female genetic reprogramming.

The regulatory relationship between *Dmrt1* and *Sox9* requires further discussion. At the sex determination stage of the mouse (E11.5), both *Sox9* and *Dmrt1* are expressed in the early embryonic testis (*Kent et al., 1996*; *Raymond et al., 1999*), but whereas early embryonic *Sox9* mutants show sex reversal (*Chaboissier et al., 2004*; *Barrionuevo et al., 2006*), early embryonic *Dmrt1* KO mice have testes that express *Sox9* and appear histologically normal until P7 (*Raymond et al., 2000*). Thus, *Sox9* expression is independent of DMRT1 during sex determination and some time thereafter. Similarly, Sertoli cell-specific inactivation of *Sox9/8* at E13.5, shortly after the sex determination stage, leads to a rapid downregulation of *Dmrt1* that becomes already visible four days later, at E17.5 (*Georg et al., 2012*). In contrast, *Dmrt1* ablation at E13.5 results in a very delayed *Sox9*

downregulation, which is seen at P14 (one month later), coinciding with *Foxl2* upregulation (*Matson et al., 2011a*). This suggests again that *Sox9* expression is independent of *Dmrt1* in newly differentiated Sertoli cells and that the loss of *Sox9* after *Dmrt1* ablation is a secondary consequence of the upregulation of ovarian genes(s), such as *Foxl2*, in the same cells. On the other hand, several observations suggest the transactivation of *SOX9* by DMRT1: 1) DMRT1 binds near the *Sox9* locus in P28 mouse testes (*Matson et al., 2011a*), 2) ectopic expression of *Dmrt1* in embryonic XX gonads causes XX sex reversal with upregulation of *Sox9* (*Zhao et al., 2015*) and 3) *FOXL2*[-/-] sex reversed polled goats undergo a process of transdifferentiation in which *DMRT1* expression precedes the upregulation of *SOX9* (*Elzaiat et al., 2014*). In the latter two cases, however, female-promoting genes, including *FOXL2*, are either downregulated or not expressed, and thus, *SOX9* upregulation could be again an indirect consequence of the downregulation of female-promoting genes. Here we provide evidence that in the adult gonad, mutant *Sox9/8* Sertoli cells lose DMRT1, and that FOXL2 protein appears concomitant with the loss of DMRT1, consistent with the notion that *Dmrt1* expression is SOX9/8-dependent and that DMTR1 represses *Foxl2*. Additional observations support this view: 1) nearly all the genes strongly affected by the loss of DMRT1 were also affected by the loss of SOX9/8; 2) Sertoli-to-granulosa cell transdifferentiation observed in the testes of our *Sox9/8* mutant mice may be reduced by decreasing levels of RA, a signaling pathway known to be blocked by DMRT1 in Sertoli cells to prevent *Foxl2* expression and transdifferentiation into granulosa-like cells (*Minkina et al., 2014*); 3) DMRT1 can silence *Foxl2* in the absence of SOX9 and SOX8 (*Lindeman et al., 2015*); and 4) *Sox9* is upregulated in the adult ovary after the ectopic expression of *Dmrt1*, coinciding with *Foxl2* downregulation (*Lindeman et al., 2015*). Altogether, available data suggest that, like at earlier stages, a main role for SOX9/8 in adult male sex maintenance is to keep *Dmrt1* actively expressed, this latter gene having a fundamental role in repressing female-specific genes. However, these observations do not rule out the possibility that DMRT1 is also necessary for the maintenance of *Sox9* expression in the adult testis and that a feed-forward regulatory loop between *Sox9/8* and *Dmrt1* exists that ensures testis maintenance and antagonizes the feminizing action of *Foxl2*. Additional experiments (e.g. a time course of *Sox9* expression in adult SC-DKO *Dmrt1* mice) will help to clarify this issue.

There is evidence that *Wt1* acts upstream of both *Sox9* and *Sox8* during the early stages of embryonic testis development (*Gao et al., 2006*; *Barrionuevo et al., 2009*). In the adult testis, we have seen that *Sox9/8*-depleted Sertoli cells initially maintain WT1 expression, but this expression becomes progressively downregulated coinciding with the time-point at which *Foxl2* is upregulated. This suggests that *Wt1* retains its hierarchical position also in the adult testis, and that female-specific factors, including *Foxl2,* may be involved in its silencing. Consistent with previous studies (*Chun et al., 1999*; *Schmidt et al., 2004*), we detected two types of granulosa cells in the normal adult ovary (*Figure 3Ce*): 1) those located in antral (mature) follicles express FOXL2 but not WT1, and 2) those located in pre-antral follicles express both proteins. Thus, considering these two molecular markers, transdifferentiation of *Sox9/8* SC-DKO Sertoli cells seems to give rise to mature follicle-type granulosa cells. This expression pattern also suggests that WT1 may play an anti-feminizing role in adult Sertoli cells.

## SOX9 and SOX8 are necessary to maintain tubular architecture and function in adult testes

We have reported here that the functional redundancy between the *Sox9* and *Sox8* alleles observed in embryonic Sertoli cells (*Barrionuevo et al., 2009*) and other embryonic cell types (*Chaboisier et al., 2004*; *Stolt et al., 2004*; *Reginensi et al., 2011*) is also maintained in adult Sertoli cells. The phenotype of mutant testes becomes ever more severe as the numbers of null alleles increase in their genotype, with extreme phenotypes observed in homozygous DKO testes 4 months after the beginning of TX treatment, at which stage seminiferous tubules have literally disappeared. As Sertoli cell proliferation stops once they obtain their adult appearance (*Kluin et al., 1984*), programmed cell death in *Sox9/8* mutants may explain their reduction in number. Consistently, we found no SOX9[+] apoptotic cell by P90 (30 datx), indicating that *Sox9/8* initially protects Sertoli cells from apoptosis, a role previously shown for this gene in other developing organs (*Akiyama et al., 2002*; *Cheung et al., 2005*; *Seymour et al., 2007*). Similarly, newly differentiated FOXL2[+] cells did also not apoptose, showing that reprogrammed granulosa-like cells are also protected from apoptosis. However, the situation was substantially different in P150 (90 datx) mutant testes, where

apoptosis was intense. At these late stages of testis regression, cord structure was dramatically compromised and even Sertoli cells still expressing *Sox9* were seen to undergo apoptosis. It is well known that the number of Sertoli cells must reach a critical threshold to organize embryonic testicular cords (*Palmer and Burgoyne, 1991*; *Schmahl and Capel, 2003*). Accordingly, our results suggest that adult testis tubules also require the presence of a minimum number of Sertoli cells to be maintained. The progressive loss of Sertoli cells after *Sox9/8* ablation, either by apoptosis or by transdifferentiation into granulosa-like cells, appears to reach a point of no return at which the remaining normal Sertoli cells are unable to support the tubular structure and are also induced to apoptose. Hence, our results show that SOXE factors are necessary to maintain Sertoli cell identity and seminiferous tubule integrity, as these cells maintain all the other cell types forming the tubules, which become completely disorganized in their absence.

Several findings suggest that deregulation of important structural proteins controlled by *SoxE* genes could be involved in the process. SOX9 controls, either directly (*Bell et al., 1997*) or indirectly (*Barrionuevo et al., 2008*; *Georg et al., 2012*), the expression of extracellular matrix proteins, which contribute importantly to the tubular structure. $Sox8^{-/-}$ mice show increased BTB permeability and greatly reduced levels of α-tubulin acetylation, suggesting that impairment of the Sertoli cell cytoskeleton may have modified the microenvironment of the seminiferous epithelium (*Singh et al., 2013*). Also, after *Sox9* ablation in *Sox8* mutants, both developing (*Barrionuevo et al., 2009*; *Georg et al., 2012*) and adult testes (present paper) experience downregulation and/or abnormal distribution of several important proteins required for the formation of Sertoli–Sertoli and/or Sertoli–germ cell adhesion complexes (*Figure 2—figure supplement 2*). In this context, it is noteworthy that spermatogenesis is halted when the functionality of the BTB is impaired (*Meng et al., 2005*; *Dadhich et al., 2013*). Thus, in our *Sox9/8* SC-DKO mouse testes, BTB permeation and cytoskeleton impairment may give rise to a damaged intra-tubular microenvironment in which spermatogenesis is not supported anymore, germ cells undergoing both apoptosis and desquamation. Altogether, available data strongly suggest that failure of *Sox9/8* double mutant Sertoli cells to sustain testis tubule architecture is a direct consequence of altered expression of cell adhesion molecules and probably of other structural elements such as components of the cytoskeleton or the extracellular matrix.

Regarding the somatic cells of the testis, PM cells disappear in *Sox9/8* mutant testes, whereas Leydig cells appear not to be affected, as they express the Leydig cell markers *HSD17b3* and *Insl3*. Although PM and Leydig cell specification is induced by Sertoli cells during early testis development (reviewed by *Svingen and Koopman, 2013*), at later stages of testis development (E14.5 and onward) Leydig cells do not require Sertoli cells for proliferation and synthesis of testosterone (*Gao et al., 2006*). Our results in the adult testis show that adult PM cells retain their original dependence from Sertoli cells, whereas maintenance of adult Leydig cells is again Sertoli cell-independent. Further research is required to unravel the actual functional status of Leydig cells in *Sox9/8* mutant testes.

## A regulatory model for adult testis maintenance in mice

According to the above considerations, we propose a model for the maintenance of Sertoli cell fate in the adult testis. In this model, *Sox9/8* play a central role in maintaining active *Dmrt1*, which prevents expression of ovary promoting genes, including *Foxl2*, which in turn negatively regulates *Sox9/8* and/or *Dmrt1*. *Dmrt1* inhibits RA signaling which promotes the expression of *Foxl2*, although an interference of *Sox9* on this signaling pathway, through a *Dmrt1*-independent mechanism, cannot be ruled out. Wt1 positively regulates *Sox8/9* and is negatively regulated by *Foxl2* and/or other ovarian-specific genes. *Sox9/8* are also needed for maintaining the expression of important testis structural genes and for protecting Sertoli cells from apoptosis (*Figure 7*, solid lines). It is also possible that *Dmrt1* may establish feed-forward regulatory loop with *Sox9/8* and that *Sox9/8* repress the expression of ovary-specific mRNAs through *Dmrt1*-independent mechanisms, although these interactions are less strongly supported by available data (*Figure 7*, dashed lines).

In conclusion, we have shown *Sox9/8* have important DMRT1-dependent and independent functions in the maintenance of the adult testis. In their absence, phenotypically normal, fertile testes are genetically reprogrammed and Sertoli-to-granulosa cell transdifferentiation occurs. Nevertheless, this is a mere transient stage of mutant adult Sertoli cells in the irreversible degenerative process the seminiferous tubules face in the absence of *Sox9* and *Sox8*.

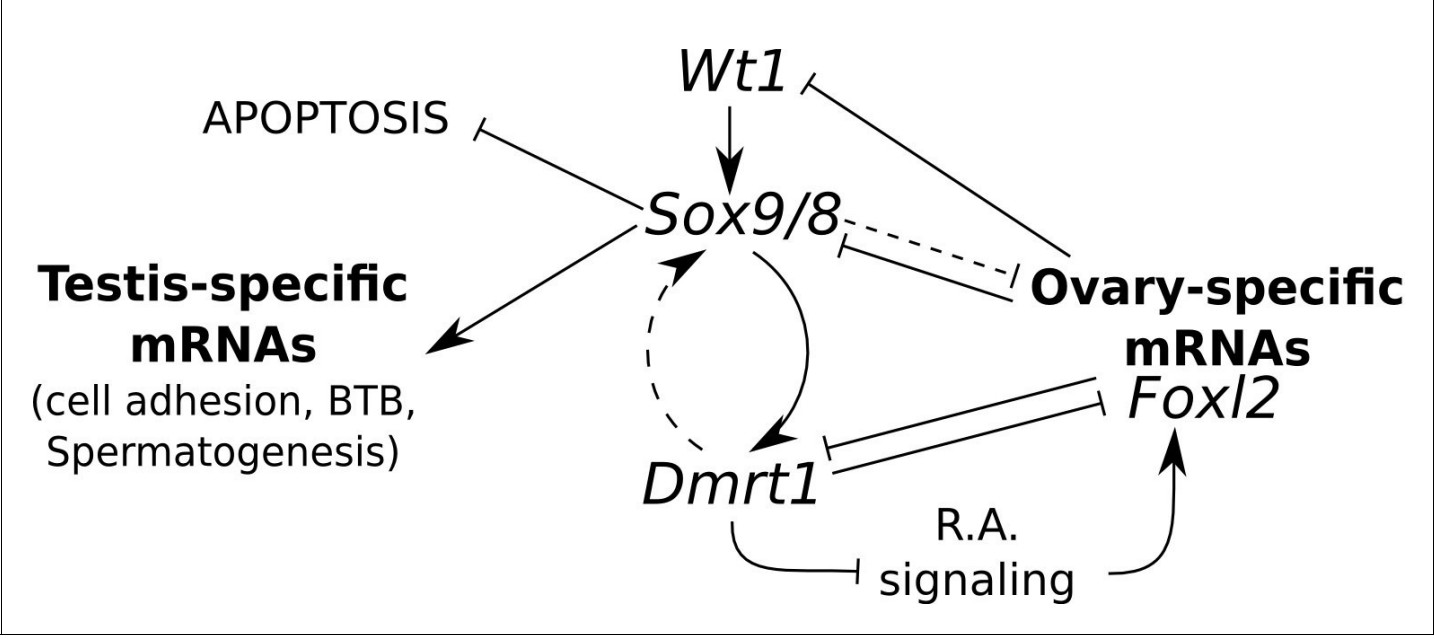

**Figure 7.** Model for the regulation of mammalian sex maintenance. Positive regulation is indicated by arrows. Negative regulation is indicated by perpendicular lines. See text for a detailed explanation.

## Material and methods

### Mouse lines and crosses

Previously generated *Sox9$^{f/f}$; Sox8$^{-/-}$* mice (*Barrionuevo et al., 2009*; *Kist et al., 2002*; *Sock et al., 2001*) were bred to *Wt1-CreERT2* mice (*Zhou et al., 2008*) and the resulting double heterozygous offspring harboring the *Cre* allele was backcrossed to *Sox9$^{f/f}$; Sox8$^{-/-}$* mice to obtain heterozygous and homozygous compound *Sox9; Sox8* conditional mutants. The same mating scheme was followed with the *Sox9-CreERT2* mouse line (*Kopp et al., 2011*). To report CRE activity, the R26R-EYFP reporter allele (*Srinivas et al., 2001*) was crossed into *Wt1-CreERT2; Sox9$^{f/f}$; Sox8$^{-/-}$ and Sox9-CreERT2; Sox9$^{f/f}$; Sox8$^{-/-}$* mice. For genotyping we performed PCR and qPCR with DNA purified from tail tips. Primers and PCR conditions for *Sox9$^{flox}$, Sox8$^{-}$, Cre,* and *R26R-EYFP* were used as described *Barrionuevo et al. (2009)*. Mouse housing and handling, as well as laboratory protocols, were approved by the University of Granada Ethics Committee for Animal Experimentation.

### Tamoxifen administration

Tamoxifen (Sigma, T5648) dissolved in corn oil (Sigma, C8267) at a concentration of 30 mg/ml and 0.16 mg of TX per gram of body weight was initially administered orally to mice with a feeding needle for 5 consecutive days. With this treatment *Sox9/8* double mutants displayed a 90% lethality, so we reduced the dose of TX (down to 0.07 mg TX / gr of body weight) and 90% of *Sox9/8* double mutants survived, but the efficiency of CRE recombination fell then to below 20%. Then, we tried to feed mice with a TX-supplemented diet (40 mg TX/100 g Harlan 2914 diet) for one month. This treatment resulted in a 100% survival rate. TX administrations were started at 2 months (P60) and finished 30 days after the beginning of TX administration (P90 [30 datx]) (*Figure 1A*). All results presented here, except those included in *Figure 1—figure supplement 2* and *3*, were obtained from mice fed with the TX-supplemented diet.

**Table 1.** Antibodies used in this study.

| Gene product | Raised in | Working dilution | References |
| --- | --- | --- | --- |
| Laminin | rabbit | 1:100 | Sigma L9393 |
| ACTA2 | mouse | 1:200 | Sigma A2547 |
| Claudin11 | rabbit | 1:100 | Santa Cruz Biotechnology, CA sc-25711 |
| DMC1 | goat | 1:100 | Santa Cruz Biotechnology, CA sc-8973 |
| PCNA | mouse | 1:100 | Santa Cruz Biotechnology, CA sc-56 |
| CYP14A1 (P450scc) | goat | 1:200 | Santa Cruz Biotechnology, CA sc-18043 |
| SOX9 | rabbit | 1:100 | Santa Cruz Biotechnology, CA sc-20095 |
| SOX9 | goat | 1:10 | Santa Cruz Biotechnology, CA sc-17341 |
| WT1 | rabbit | 1:100 | Santa Cruz Biotechnology, CA sc-192 |
| FOXL2 | goat | 1:100 | Abcam ab5096 |
| GFP | rabbit | 1:100 | Novus Biologicals NB600-308 |
| WT1 | mouse | 1:30 | DAKO M3561 (clone 6F-H2) |
| CYP19A1 (Aromatase) | mouse | 1:10 | GeneTex GTX41561 |
| DMRT1 | rabbit | 1:400 | Gift from Dr. Silvana Guioli |

## Histological and immunostaining methods

Gonads were dissected out, weighted and prepared for standard histological methods, including haematoxylin and eosin staining. Single and multiple immunofluorescence were performed as previously described (*Dadhich et al., 2013*). *Table 1* summarizes the antibodies used.

## Analysis of apoptosis

To perform the TUNEL technique we used the Fluorescent In Situ Cell Death Detection Kit (Roche, Mannheim, Germany) according to the manufacturer's instruction.

## Analysis of BTB Permeability

The in vivo test to analyze the permeability of the BTB in the testes of control and mutant mice was performed using a biotin-labelled tracer compound (EZ-Link Sulfo-NHS-LC-Biotin tracer, Thermo Scientific) as described (*Dadhich et al., 2013*).

## WIN 18,446 treatment

TX-treated *Sox9-CreERT2; Sox9$^{f/f}$; Sox8$^{-/-}$* mice were injected subcutaneously either with 40 µg/µl WIN 18,446 (Tocris, Biotechne, UK, Cat. No 4736), dissolved in 50 µl dimethyl sulfoxide or with the vehicle alone for 8 days, 4 days before and 4 days after the end of the 30 days diet TX treatment. Fifteen days after the end of WIN 18,446 treatment, gonads were collected and processed for double immunofluorescense for ACTA2 and FOXL2 as described above. The number of FOXL2$^+$ cells per transversal ST section was counted in 20 tubules of 5 WIN 18,446-treated and 5 control animals. Only circular or ellipsoid tubular sections in which the major/minor axis ratio was lower than two were used for counts.

## Transcriptome analysis

Both testes were extracted from six P150 (90 datx) mutant males (three *Wt1-CreERT2; Sox9$^{f/f}$; Sox8$^{-/-}$* and three *Sox9-CreERT2; Sox9$^{f/f}$; Sox8$^{-/-}$*). As controls, both gonads were also extracted from two P150 (not treated) and two P150 (90 datx) *Sox9$^{f/f}$* male mice as well as from two 4–5 months old normal females. All TX-treated mice were euthanized three months after the initiation of diet TX-treatment for one month. The two gonads of each individual were pooled, homogenized in 1 ml of RNAzol (Molecular Research Center, Inc.) per 100 mg of tissue and the total RNAs were then individually purified from the twelve samples following the RNAzol

manufacturer's instructions. After successfully passing Macrogen Inc. quality check, the twelve RNAs were paired-end sequenced separately in an Illumina HiSeq 2000 platform at that company and the quality of the resulting sequencing reads was assessed using FastQC (http://www.bioin-formatics.bbsrc.ac.uk/projects/fastqc/).

## Bioinformatics

RNAseq data were processed with the Tuxedo tools (*Trapnell et al., 2012*). Alignments were done with Tophat/Bowtie2 against the mm10 UCSC annotated mouse genome. Differential expression analyses where done with Cuffdiff. Analysis of the resulting data were performed with the CummeR-bund Bioconductor package. The quality of RNA-seq was checked as described in the package documentation. Briefly, by comparing FPKM scores across samples, and looking for outliers replicates, by analyzing squared coefficient of variation which allows visualization of cross-replicate variability between conditions and by analyzing the dispersion plots (*Figure 8*).

To explain the presence of both *Sox9* and *Sox8* transcripts in the transcriptome of double homozygotes for the null allele, their transcripts where visualised with the IGV genome browser (*Robinson et al., 2011*). Recombinant *Sox9* locus is seen by Cuffkinks and IGV as an alternative spliced transcript. Sashimi plots show that the CRE recombination is not 100% effective as transcripts with the correct splicing still remain in both mutant conditions but a high proportion of the *Sox9* genes are efficiently deleted. These plots also show that in the absence of the 2nd and 3rd exons after recombination, alternative intron donor and acceptor sites downstream of *Sox9* can be used

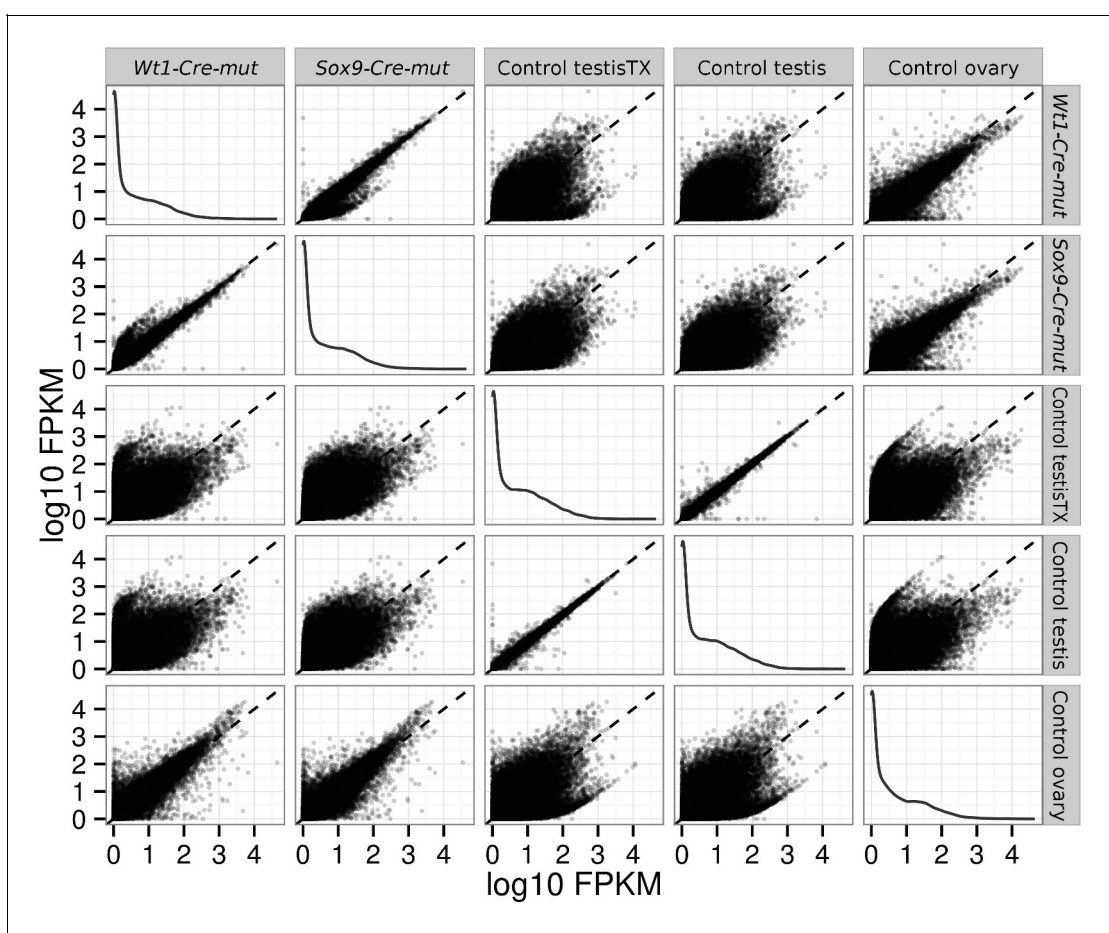

**Figure 8.** Pairwise scatterplots comparing log$_{10}$ FPKM between different conditions. Dispersion is lower when comparing similar conditions (controls, mutants) and higher when comparing mutant with control conditions. Notice that dispersion observed when mutants are compared with ovary is lower than that observed comparing them with any of the testis controls.

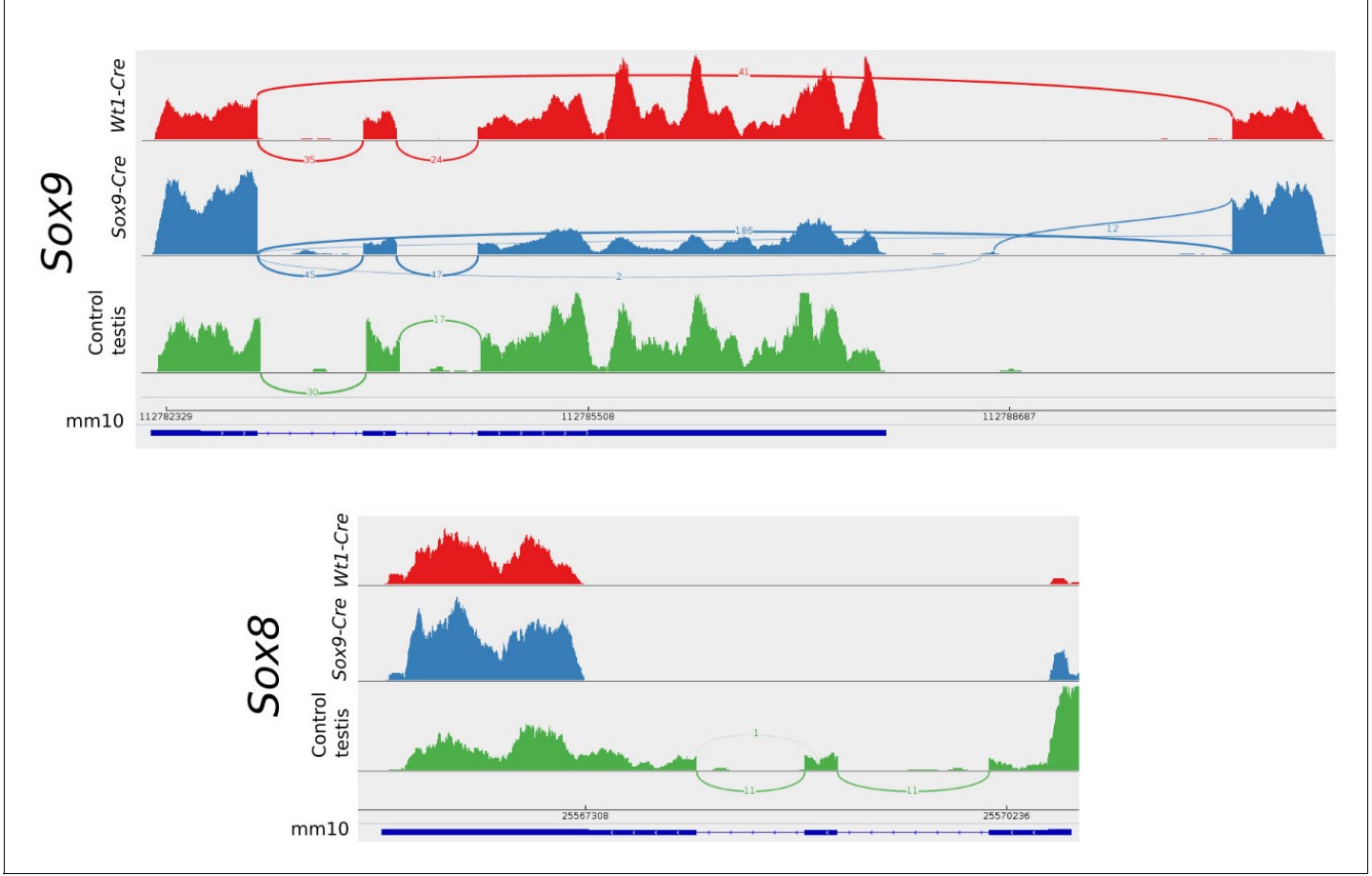

**Figure 9.** Sashimi plots of the *Sox9* and *Sox8* genes in mutant and control conditions. Vertical lines indicate coverage and curved lines indicate splicing. The mm10 row shows the positions of the exons and introns and the translated region as annotated in the mm10 UCSC mouse genome. Aberrant splicing sites where found in mutant but not in control samples. Notice that normal splicing also occurred in mutant animals showing that CRE-recombination efficiency was not 100%.

for splicing. *Sox8* transcripts only include the 5' untranslated portion of the transcripts demonstrating that these individuals are actually Sox8$^{-/-}$ (*Figure 9*).

For *Sox9/8* DKO and *Dmrt1* KO transcriptome comparison CEL files corresponding to the *Dmrt1* conditional knockout expression analysis of P28 testes by *Matson et al. (2011a)* were downloaded from the GEO database (Acc: GSE27261). Files were processed with the simpleaffy (*Miller, 2016*) package from Bioconductor and normalized with gcrma (*Wu and Gentry, 2016*). Uninformative data, control probes and genes with low variation or close to background were filtered out. Data were grouped in two conditions, Control and Mutant. Differential expression was analyzed with the limma package (*Richie et al., 2015*) and annotated with the Affymetrix Mouse Genome 430 2.0 Array annotation data. Genes with log$_2$FC having p values less than 0.05 for differential expression tests respect to normal testes where selected. These genes list was then selected from our transcriptome data and those showing non-significative log$_2$FC where filtered out. The remaining 8910 genes showing significant differential expression in both experiments are included in *Figure 5—source data 1*.

## Acknowledgements

This work was supported by grants from the Andalussian Government, Junta de Andalucía, (BIO-109) and grant P11-CVI-7291 to M Burgos, the Spanish Ministry Science and Innovation (CGL2011-23368) to R Jiménez, grant BFU2010-16438 to M Bakkali, grants from the German Research

Foundation to G Scherer (DFG Sche 194/18-1, GRK 1104) and grants NIH-NIDDK DK078803 and DK068471 to M Sander. The authors would like to thank Dr. M Wegner for contributing the *Sox8* mutant mouse line, Dr. S Guioli for kindly providing us with the DMRT1 antibody, S Kaltenbach for help with mouse husbandry in Freiburg and the Spanish Ministerio de Ciencia e Innovación for the 'Ramón y Cajal' fellowship granted to M Bakkali and the PhD fellowship granted to A Hurtado.

## Additional information

### Funding

| Funder | Grant reference number | Author |
|---|---|---|
| Ministerio de Ciencia e Innovación | BFU2010-16438 | Mohammed Bakkali |
| National Institute of Diabetes and Digestive and Kidney Diseases | DK078803 | Maike Sander |
| National Institute of Diabetes and Digestive and Kidney Diseases | DK068471 | Maike Sander |
| Deutsche Forschungsgemeinschaft | DFG Sche 194/18-1 | Gerd Scherer |
| Deutsche Forschungsgemeinschaft | GRK 1104 | Gerd Scherer |
| Andalussian Government, Junta de Andalucía | P11-CVI-7291 | Miguel Burgos |
| Ministerio de Ciencia e Innovación | CGL2011-23368 | Rafael Jiménez |
| Andalussian Government, Junta de Andalucía | BIO109 | Rafael Jiménez |

The funders had no role in study design, data collection and interpretation, or the decision to submit the work for publication.

### Author contributions

FJB, MBu, RJ, Conception and design, Acquisition of data, Analysis and interpretation of data, Drafting or revising the article; AH, G-JK, FMR, Acquisition of data, Analysis and interpretation of data; MBa, This authors performed the initial bioinformatic analysis of the transcriptome data, Acquisition of data; JLK, Acquisition of data, Contributed unpublished essential data or reagents; MS, Drafting or revising the article, Contributed unpublished essential data or reagents; GS, Conception and design, Drafting or revising the article

### Author ORCIDs

Francisco J Barrionuevo, http://orcid.org/0000-0003-2651-1530
Mohammed Bakkali, http://orcid.org/0000-0002-9079-8452
Janel L Kopp, http://orcid.org/0000-0002-1875-3401
Miguel Burgos, http://orcid.org/0000-0003-4446-9313
Rafael Jiménez, http://orcid.org/0000-0003-4103-8219

### Ethics

Animal experimentation: This study was performed in strict accordance with the guidelines for the protection of the animals used in scientific experimentation (Decree-Law 53/2013), dictated by the Spanish Ministry of Presidency. The protocol was approved by the Ethical Committee for Animal Experimentation of the University of Granada (Ref. No.: 123-CEEA-UGR-2011). All surgery, except the BTB permeability experiment, was performed post-mortem after cervical dislocation. BTB experiment was performed under anesthesia for 30 min and then the animals were sacrificed without recovery. Every effort was made to minimize suffering.

# Additional files

## Supplementary files

• Supplementary file 1. Complete lists of genes included in the 8 molecular pathways mentioned in *Figure 2—figure supplement 2*.

## Major datasets

The following dataset was generated:

| Author(s) | Year | Dataset title | Dataset URL | Database, license, and accessibility information |
|---|---|---|---|---|
| Barrionuevo FJ, Hurtado A, Kim GJ, Real FM, Bakkali M, Kopp JL, Sander M, Scherer G, Burgos M, Jimenez R | 2016 | Testis-to-ovary reprogramming and testis regression after ablation of Sox9 in Sox8-/- mice | https://www.ebi.ac.uk/arrayexpress/experiments/E-MTAB-4572/ | Publicly available at the European Bioinformatics Institute Array Express (Accession no: E-MTAB-4572) |

The following previously published dataset was used:

| Author(s) | Year | Dataset title | Dataset URL | Database, license, and accessibility information |
|---|---|---|---|---|
| Matson CK, Murphy MW, Sarver AL, Griswold MD, Bardwell VJ, Zarkower D. | 2011 | Dmrt1 (doublesex and mab-3 related transcription factor 1) conditional knockout expression analysis of P28 testes | http://www.ncbi.nlm.nih.gov/geo/query/acc.cgi?acc=GSE27261 | Publicly available at the NCBI Gene Expression Omnibus (Accession no: GSE27261) |

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
