## [Decision Letter]

Thank you for submitting your article "*Sox9* and *Sox8* protect the adult testis from male-to-female genetic reprogramming and complete degeneration" for consideration by *eLife*. Your article has been reviewed by three peer reviewers, and the evaluation has been overseen by Janet Rossant as the Senior Editor. One of the three reviewers has agreed to reveal his identity: Steven Munger (Reviewer #1).

The reviewers have discussed the reviews with one another and the Reviewing Editor has drafted this decision to help you prepare a revised submission.

Summary:

Over the past few years, a new concept appears in reproductive biology showing that the sexual fate of adult supporting cells in male and female gonads is not fully determined and that transdifferentiation of granulosa to Sertoli can occur. The effect of adult knock-out of both *Sox9* and SOX8 was clearly awaited, given the known key roles of these genes in establishing testis fate in development. Overall this manuscript reports a tremendous amount of work with the characterization of mutant gonads at many different time-points after P60 by histology, immunohistochemistry using many relevant cell markers, and transcriptomic analyses by RNA-sequencing at one developmental stage. The results are detailed, well illustrated, well reported and convincing. The results presented here are novel and are of broad interest in understanding transdifferentiation and the stability of adult differentiated cell types

Essential revisions:

The reviewers raised a few essential points that need to be addressed in a revised manuscript:

1) Better explanation of TX administration and timeline; inclusion of diagram to outline dosage details. Address potential indirect reproductive effects from TX.

2) Include Excel table of differential expression comparison from RNA-seq.

3) Moderate any conclusions made on the basis of weak double staining in confocal images.

These are detailed below.

1) In the present study, authors have mainly characterized testes after a 3 months exposure to tamoxifen (introduced in the food as stated in the subsection “Tamoxifen administration”), stage named P150 (90 datx). By contrast, in an equivalent study on *Dmrt1*, Matson and colleagues treated P60 mice, carrying the UbiquitinC-cre/ERT2, by only two IP-injections of 4mg tamoxifen/mouse (Matson et al., 2011, Nature 476:101-4). And surprisingly here, after the 3-months TX-treatment the *Sox9* gene is far to be completely deleted as presented on Supplemental Materials and methods, first paragraph (with apparently a better efficiency of the *Sox9-Cre* compare with the Wt1-cre; fitting with results presented in the first paragraph of the Results section). Do the authors have an idea to explain this poor efficiency of this TX-treatment? And how are they sure to study double KO *Sox8/9* testes?

As tamoxifen is a selective estrogen-receptor modulator, its action per se remains questionable, especially on reproductive organs. A negative answer to this question could emerge from transcriptomic data (Figure 2, as example) where TX-treated control and control testis present the same transcriptomic profile. But, uncertainty remains for TX-treatment and an eventual discrepancy between control and mutant animals. As example: it remains difficult to understand if a stage named P150 (90datx) corresponds (i) to a mouse with tamoxifen during 90 days after P60 or (ii) with tamoxifen during 30 days (from P60 to P90) then without tamoxifen from P90 to P150. In the latter case it has been better to note the stage P150 (30 datx + 60) instead of P150 (90 datx). Confusion about this arises in subsection “Transcriptome analysis” compared with the subsection “Somatic testis-to-ovary genetic reprogramming in the absence of *Sox9* and *Sox8* in adult mouse testes”; and also from sentences in the subsection “Tamoxifen administration” compared with the legend to Figure 1—figure supplement 2 ("P150 (60 datx)"); and also from the first paragraph of the Results section. Authors should clarify this point all along the text, the figures and the legends of the figures. Moreover, all along the manuscript authors write "TX-treated controls" instead of writing "PXX (ZZ datx)-control" where XX will represent the stage of observation (from 70 to 180) and ZZ the duration of TX-treatment. Authors should indicate in each case how long TX has been administered to control animals. Authors should also indicate the genotype of the control. Normally, the best control of a *Sox9^flx/flx^;Sox8^-/-^;Sox9-CreERT2*-P150 (90 datx) will be a *Sox9^flx/flx^;Sox8^-/-^*;Control-P150 (90 datx). Is it the case?

Figure 5 – One concern I have is that the authors may be over interpreting the observed positive linear trend in their comparison of differentially expressed gonad genes from their current (RNA-seq based) *Sox9/8* dco study at P150 to an earlier published microarray analysis of SC-*Dmrt1*KO testes at P28 (Matson et al. 2011). The authors conclude that *Dmrt1* and *Sox9/8* likely act in the same pathway because nearly all genes strongly affected by loss of *Dmrt1* were also affected by the loss of *Sox9/8*. Although I agree this is likely to be true, this pattern of gene expression could be a secondary effect of the change in relative numbers of cell types in these whole gonad samples. Any gene KO that results in loss of Sertoli cells will likely cause this change in the observed expression pattern of known sex-determining genes regardless of which specific pathway they are affecting in the Sertoli cell.

2) This was requested by one of the reviewers to assist with analysis of data.

3) Figure 4 also shows double "weak staining" and in contrast with Figure 3, Figure 4 provides only merge for SOX9 and FOXL2, the single colors for each staining should be added to convince the reader. Same observation for DMRT1/FOXL2 double staining (Figure 4 b and b'). These double staining are the weak point of the manuscript because the authors make from these experiments some important conclusions. It would be helpful for the reader if the authors could bring some counting of these "double stained cells" to support their conclusions.

---

## [Author Response]

*Essential revisions:*

The reviewers raised a few essential points that need to be addressed in a revised manuscript:

*1) Better explanation of TX administration and timeline; inclusion of diagram to outline dosage details. Address potential indirect reproductive effects from TX.*

The reviewers were concerned about four main issues:

A) The administration of the tamoxifen (TX) treatment and the notation we used to refer to the analyzed stages;

B) The CRE recombination efficiency;

C) The possible collateral effects that TX could have had on the phenotype of our control and mutant mice;

D) Our interpretation of the results when comparing our RNA-seq data with those from the microarray analysis of SC-*Dmrt1*KO P28 testes performed by Matson et al. (2011).

A) Regarding the tamoxifen treatment, the reviewers stated that the “authors have mainly characterized testes after a 3 months exposure to tamoxifen”.

This is a clear confusion as in the Results section and Materials and methods section of our first version of the manuscript it was indicated that diet treatment lasted for 30 days/one month. We initially treated adult mice with TX at a concentration of 30mg/ml and 0.16 mg of TX per gram of body mass using a feeding needle for 5 consecutive days. However, *Sox9/8* double mutants displayed 90% lethality with this treatment. This is probably related to the fact that *Sox8^-/-^* mice display an idiopathic body mass reduction and are weaker than WT ones, as reported previously (Sock et al., 2001. Idiopathic weight reduction in mice deficient in the high-mobility-group transcription factor *Sox8*. Mol Cell Biol. 21: 6951–6959). Then, we reduced the dose of TX (down to 0.07 mg TX / gr of body mass) and 90% of *Sox9/8* double mutants survived, but the efficiency of CRE recombination in adult Sertoli cells fell then to below 20%. Since acute/short TX treatments were clearly useless in our animal model, we then tried to feed mice with a TX-supplemented diet (40mg TX/100g Harlan 2914 diet) for one month (never for three months).

TX administration always started at 2 months (postnatal stage P60) and finished just 30 days later (at P90). Thus, a stage named “P150 (90 datx)” (datx: days after TX treatment initiation) corresponds to a mouse treated for 30 days with TX-supplemented diet (from P60 to P90) and fed afterwards with normal diet, from P90 to P150. This treatment resulted in a 100% survival rate and permitted us to induce outstanding phenotypes in the mutant animals. So, all the data presented in the manuscript, except those shown in Figure 1—figure supplement 2 and 3 (see below for an explanation), were obtained from mice treated with the TX-supplemented diet as described (one month).

We assume that, like the reviewers, any other reader could also be confused regarding TX-treatment and admit that our ms was not clear enough in this respect. For this reason, as suggested by the editors, we have included a diagram in Figure 1 (the new Figure 1) illustrating the experimental schedule of the TX treatment and explaining how we have noted the main stages analyzed in the study. In our opinion, including this information at the very beginning of the Results section will ensure to make this point definitely clear. Hence, we believe that our original notation of the analyzed stages [P150 (90datx), for instance] will now be easily understandable for the reader, so we decided to keep it as it stands. Nevertheless, we would change the notation as suggested by the reviewers [P150 (30 datx + 60), for instance] if the editor/reviewers still consider that this system is better.

The reviewers also had some concerns regarding the TX-treated controls. Control animals shared the same cages and diet with mutant ones, so both control and mutant mice received exactly the same treatment. So, we do not see the need and convenience of using a different notation for them. The treatment of control animals was always indicated in the figures and in the figure legends. For instance, in the frame at the top of Figure 1Bb (the new one) it is simply noted Control, but in another frame placed just below it is noted the stage [P150 (90 datx)] which evidences that both control and mutants were treated with TX in the same way. The same system was used in all figures. In addition to the mutant samples, we always placed testis sections from four control animals in our IFs and IHCs histological preparations: WT (both TX-treated and untreated) and *Sox9^f/f^;Sox8^-/-^* (both TX-treated and untreated). Notably, we never found a significant difference between them in the expression pattern of any of the markers we analyzed in this study. Moreover, the micrographs we selected to be included in the figures were always from the TX-treated *Sox9^f/f^;Sox8^-/-^* mice. Nevertheless, following the reviewers’ recommendations, we have included in the figure legends the genotype of every control mouse that was not explicitly mentioned in the text.

Similarly, changes have been made in several points of the text (subsection “Somatic testis-to-ovary genetic reprogramming in the absence of *Sox9* and *Sox8* in adult mouse testes”, first paragraph, subsection “Tamoxifen administration” and subsection “Transcriptome analysis”), for which the reviewers reported some confusion on TX treatment.

Figure 1—figure supplement 2 and 3 report data from the only experiments in which mice were not treated with a TX-supplemented diet (see above). Instead, TX was administered orally with a feeding-gauge needle. We started using the WT1-CRE line to generate the SC *Sox9/8* DKO mice, and TX was initially administered either intraperitoneally or with a feeding-gauge needle (we obtained better results with the feeding needle). All control and mutant mice with compound homozygous and heterozygous genotypes containing less than 2 *Sox8* null alleles (either 0 or 1) tolerated well the TX treatment, but those containing 2 *Sox8* null alleles exhibited high lethality, permitting us to analyze only some few surviving animals. Nevertheless, the fact is that the results obtained in those initial studies clearly show the redundancy between *Sox9* and *Sox8* in adult testis maintenance, and we considered unnecessary and inappropriate to sacrifice so many additional animals for simply repeating the same experiment using the TX-supplemented diet. This is why we included the following sentence in our first version of the manuscript: “All results presented here, except those included in Figure 1—figure supplement 2 and Figure 1—figure supplement 3, were obtained from mice fed with the TX-supplemented diet.”

According to these considerations, and in order to avoid any confusion in the rest of the manuscript, we have made changes in the text as shown in legends of Figure 1 and Figure 1—figure supplement 1.

The reviewers also expressed some concern regarding the sentence in the first paragraph of the Results section, but we believe that with the above changes this sentence is actually not confusing in the new version.

B) Regarding TX, the reviewers had another concern related to the presumed low efficiency of our treatment system, compared with that used by Matson et al. (2011, Nature 476:101-104).

As far as we know, no paper has reported to date a TX-induced CRE recombination in adult mouse Sertoli cells showing an efficiency higher than the one we present here. Matson and col. used Ubc-CRE/ERT2 mice to delete *Dmrt1* in adult mice and they observed that after TX injection DMRT1 expression declined rapidly in germ cells while remained constant in Sertoli cells over the following 8 days (Matson et al., 2010) The mammalian doublesex homolog DMRT1 is a transcriptional gatekeeper that controls the mitosis versus meiosis decision in male germ cells. Dev. Cell 19, 612–624]. In the 2011 work, they used the same mice and the same treatment, and they were able to show just a few (only 4) FOXL2+ cells within a testis cord one month after TX administration, evidencing a very low recombination efficiency in Sertoli cells. Actually, no data about the efficiency or kinetics of CRE recombination in Sertoli cells was reported in that paper. Other studies have also revealed that TX-induction of CRE-mediated recombination in adult Sertoli cells it is not an easy task with the available tools. For example, conditional inactivation of Wt1 using a CAG-CRE, revealed efficient recombination in kidney, ovary, pancreas, spleen, lung, and uterus, but not in Sertoli cells (Chau, You-Ying et al., 2011). Acute multiple organ failure in adult mice deleted for the developmental regulator Wt1. PLoS Genet 7.12: e1002404]. Here we show that the number of SOX9+ cells per seminiferous tubule cross section decreased to 37% in the testes of the *Wt1- Cre-ERT2; Sox9^f/f^;Sox8^-/-^* mice and to 69% in the *Sox9-Cre-ERT2;Sox9^f/f^; Sox8^-/-^* testes when compared to controls (new Figure 1). This reduction is even more pronounced if we compare the Sashimi plots for *Sox9* obtained from our transcriptome data (see supplemental Materials and methods), which show a reduction of the WT *Sox9* transcripts of 57% and 80% for the *Wt1-Cre* DKO and the *Sox9-Cre* DKO lines, respectively. Thus, we disagree with the notion that we managed with low CRE recombination efficiency in our experiments. Rather, we believe we were able to develop a TX administration schedule that provides the highest CRE recombination efficiency reported to date in Sertoli cells (demonstrated with two different mouse CRE lines). We consider that this method could be a valuable tool for future gene targeting studies in adult Sertoli cells.

Based on the presumed low efficiency of CRE recombination, the reviewers asked: “how are they (the authors) sure to study double KO *Sox8/9* testes”? We are absolutely sure we studied double KO *Sox8/9* testes for two reasons:

1) We have used a *Sox8^-/-^* mouse in which the WT SOX8 protein is absent, as reported previously (Sock et al., 2001) Idiopathic weight reduction in mice deficient in the high-mobility-group transcription factor *Sox8*. Mol Cell Biol. 21: 6951–6959]. Consistent with this, the Sashimi plots for *Sox8* obtained from our transcriptome data (see Supplemental Materials and methods) show that in the *Sox8^-/-^* mice, *Sox8* transcripts only conserve the 5’ untranslated region, demonstrating that these individuals are actually homozygous for a *Sox8* null allele.

2) The Sashimi plots for *Sox9* confirm that, as expected according to the features of the *Sox9^flox^* line we used (Kist et al. (2002). Conditional inactivation of *Sox9*: a mouse model for campomelic dysplasia. Genesis 32: 121–123], CRE recombination generates *Sox9* transcripts lacking the 2nd and 3rd exons. In addition, our IF studies demonstrated that the SOX9 protein is highly reduced in the SC *Sox9/8* adult testes after TX administration (Figure 1; Figure 1—figure supplement 1).

C) “Address potential indirect reproductive effects from Tx”.

Two previous studies have investigated the effect of TX on male fertility on rats: 1) Gopalkrishnan, et al. (1998) Tamoxifen-induced light and electron microscopic changes in the rat testicular morphology and serum hormonal profile of reproductive hormones. Contraception, 57: 261-269; and 2) Gill-Sharma et al., (1993) Effects of tamoxifen on the fertility of male rats. J. Reprod. Fertil. 99: 395-402. In these studies, TX was administered during 90 days at different concentrations: 0.04, 0.2 and 0.4 mg TX /gr body weight/day. The authors observed a progressive degeneration of the germinative epithelium already detectable by day 10 after TX administration. By day 50, the phenotype was quite severe, with prominent reduction of the germinative epithelium thickness and low numbers of spermatids and spermatozoa. The severity of this phenotype increased with the TX concentration.

Importantly, they also showed that this reduced fertility was completely restored 90 days after drug withdrawal. We have administered to our mice a dosage equivalent to the lowest used in these studies (0.04 TX /gr body weight/ day). In fact, we supplemented our diet with 40 mg TX/100 mg food; a 25 mg mouse eats around 3 gr of food per day and our mice were fed only for 30 days with this diet. We described the effect of this concentration of TX:: “TX-treated controls were similar to untreated males, except between P80 (20 datx) and P120 (60 datx) and mainly at P90 (30 datx), when they showed some degenerating seminiferous tubules, but recovered afterwards”. Additional related information is provided in Figure 1 and Figure 1—figure supplement 4.

Indeed, at the histological level, P90 (30 datx) testes showed degenerating seminiferous tubules, and the number of spermatozoa was reduced, but at P120 (60 datx) control testes showed almost no difference with the testes of P120 untreated males. At P150 we found no difference between treated (90 datx) and untreated testes. At this later stage, transcriptome comparison showed no difference between TX-treated and untreated testes. In addition, we found no difference in the expression pattern of several proteins analyzed by IHC and IF between TX-treated and untreated controls at any stage.

Thus, we are confident that TX treatment did not affect the results of our study on *Sox9/8* mutant mice.

D) Another concern of the reviewers was that “the authors may be over interpreting the observed positive linear trend in their comparison of differentially expressed gonad genes from their current (RNA-seq based) *Sox9/8* dco study at P150 to an earlier published microarray analysis of SC-*Dmrt1*KO testes at P28 (Matson et al. 2011)”.

We believe the reviewers are correct. A possibility remains that the coincident gene expression pattern observed in these two profiling studies could also have been a secondary effect of the changes in the relative numbers of cell types occurred in both cases. Accordingly, we have changed the text in order to include this possibility (see subsection “Sertoli-to-granulosa cell transdifferentiation is mediated by *Dmrt1* downregulation in *Sox9/8* SC-DKO testes”, second paragraph). We observed that nearly all genes strongly affected by the loss of *Dmrt1* were also affected by the loss of *Sox9/8*, with a regression line whose scope is nearly 1. Based on this, we stated that these results suggest that “both *Dmrt1* and *Sox9/8* act in the same pathway”. But this is just another finding in support of this hypothesis, not the only one. Additional supporting evidence exist: 1) in both *Sox9/8* and *Dmrt1* mutants, *Foxl2* is upregulated; 2) *Sox9/8* ablation leads to *Dmrt1* downregulation; 3) *Dmrt1* ablation leads to *Sox9/8* downregulation; and 4) in both *Sox9/8* and *Dmrt1* mutants, RA-signaling pathway is affected. Overall, available evidence supports the notion that these two genes act in the same pathway in Sertoli cells.

2) Include Excel table of differential expression comparison from RNA-seq.

We have included the differential expression comparison from RNA-seq in [Supplementary-material SD2-data], we made a call to this file in the subsection “Somatic testis-to-ovary genetic reprogramming in the absence of *Sox9* and *Sox8* in adult mouse testes”, and we have included a legend.

*3) Moderate any conclusions made on the basis of weak double staining in confocal images.*

The single color channels for Figure 4 were already included in Figure 4—figure supplement 1 of the first version of the manuscript. In a former revised version of this manuscript submitted to another journal, we composed a different Figure 4 including the single color channels for Figure 4. However, one of the reviewers’ concerns was that the resulting figure was overwhelming. That is why in this version of the manuscript we decided to include the single channels in a supplemental figure. As suggested by the *eLife* reviewers, we have counted 203 FOXL2+ cells from 4 animals at P120 (60 datx) and 12 of them coexpressed SOX9. Likewise, we counted 127 FOXL2+ cells from 4 animals at P120 (60 datx) and 16 coexpressed DMRT1. We changed the text accordingly in the first paragraph of the subsection “Sertoli-to-granulosa cell transdifferentiation is mediated by *Dmrt1* downregulation in *Sox9/8* SC-DKO testes”.